# Towards Provably Personalized Federated Learning via Threshold-Clustering of Similar Clients

## Abstract

Clustering clients with similar objectives together and learning a model per cluster is an intuitive and interpretable approach to personalization in federated learning (PFL). However, doing so with provable and optimal guarantees has remained an open challenge. In this work, we formalize personalized federated learning as a stochastic optimization problem where the stochastic gradients on a client may correspond to one of $K$ distributions. In such a setting, we show that using i) a simple thresholding based clustering algorithm, and ii) local client momentum obtains optimal convergence guarantees. In fact, our rates asymptotically match those obtained if we knew the true underlying clustering of the clients. Further, we extend our algorithm to the decentralized setting where each node performs clustering using itself as the center.

## 1 Introduction

We focus on the cross-silo Federated learning setup which allows $N$ clients to collaborate on training a global model without exchanging their raw data [6]. The defacto algorithm for this setup is federated averaging [9], in which each client $i \in [N]$ has a loss function $f_i(x)$ which they wish to minimize. At time step $t$, each client $i$ performs $m$ local stochastic updates starting from the current parameters of a global model $y_{i,t}^0 = x_{t-1}$

$$y_t^l = y_{i,t}^{l-1} - \eta g_i(y_{i,t}^{l-1}) \quad \text{for } l \in \{1, \ldots, m\}.$$

These updates are then sent to the central server who averages them to obtain the new centralized model as below and broadcasts the updated parameters back to the clients

$$x_t = \frac{1}{N} \sum_{i=1}^{N} y_t^m,$$

While this framework preserves data privacy (since it only requires exchanging gradients, not raw data) and streamlines communication via a central coordinator, it averages the gradients of *all* clients for each parameter update, which might be undesirable due to heterogeneity of data distributions across clients [12, 13]. Ideally, the model would be adaptive to different subsets of similar clients. That is, only clients with similar data distributions would learn from each other and share updates, unaffected by dis-similar clients [11]. To address this, we propose a *personalized* federated learning algorithm, which simultaneously clusters similar clients and optimizes their loss objectives in a personalized manner. Before each parameter update, the server first clusters similar clients and then computes personalized parameter updates based only on the gradients of clients within that cluster.

**Our contributions.**
- We propose two personalized learning algorithms (one federated and one decentralized) which converge at the optimal $\mathcal{O}(1/\sqrt{n_i T})$ rate for stochastic gradient descent for non-convex functions where $n_i$ is the number of clients similar to client $i$.

---

**Algorithm 1** Threshold-Clustering

---

**Input** Points $\{x_1, ..., x_N\}$; Number of clusters $K$; Initializations of cluster-mean estimates $\{v_{1,0}, ..., v_{K,0}\}$

1: **for** round $l \in [M]$ **do**
2:     **for** cluster $k$ in [K] **do**
3:         Set threshold $\tau_{k,l}$.
4:         Update cluster-mean estimate:

$$v_{k,l} = \frac{1}{n} \sum_{i=1}^{N} \left( x_i \mathbb{1}(\|x_i - v_{k,l-1}\| \leq \tau_{k,l}) + v_{k,l-1} \mathbb{1}(\|x_i - v_{k,l-1}\| > \tau_{k,l}) \right). \quad (1)$$

      **return** Cluster-mean estimates $\{v_k = v_{k,M}\}_{k \in [K]}$ and clusters $\{\mathcal{C}_k = \{x_i : \|x_i - v_{k,M}\| \leq \tau_{k,M}\}\}_{k \in [K]}$

---

- We introduce a novel clustering subroutine based on thresholding, whose performance improves with the separation between the cluster means and the number of data points being clustered. We also prove nearly matching lower bounds showing its near-optimality.
- We show experimentally that our clustering algorithm benefits from collaborative learning, is competitive with SOTA personalized federated learning algorithms, and is not sensitive to the model's initial weights.

**Related work.** Personalization in federated learning has recently enjoyed tremendous attention and we refer to [11] for a survey. Of these, [5, 4, 8, 10] have considered clustering methods. Closest to our approach are [5], who propose a general framework for cluster-based personalized FL that supports both gradient and model averaging. They require that all loss objectives be strongly convex whereas our theory requires only smoothness of the loss objectives. With *HypCluster*, [8] train a global model that has strong generalization guarantees. Features of their algorithm are somewhat orthogonal to the privacy-preserving spirit of federated learning, as they assume the server can access the clients' raw data. The algorithm we design to cluster similar clients is closely inspired by the clipping procedure in [7] for byzantine robust optimization, which distinguishes 'good' from 'malicious' clients by iteratively updating an estimate of the mean of the good clients.

## 2 Thresholding-based Clustering

In order to personalize models for different clients, we propose a robust clustering procedure which groups similar clients together. Our algorithm (Algorithm 1) is simple: given a set of points $\{x_1, ..., x_N\}$ which can be partitioned into $K$ clusters of identically distributed points, we set initial estimates of the cluster-centers, $\{v_{k,0}\}_{k \in [K]}$. Then, for each cluster-center estimate, we assign a value to every point in the dataset: **(a)** if a point is near the cluster-center estimate, we preserve its value; **(b)** if a point is far away, we set its value to the cluster-center estimate. Finally, to compute the updated cluster-center estimate, we average the values of all the points (see update rule (1)).

Assignment rule **(a)** ensures that cluster-center updates are influenced by nearby points. Assignment rule **(b)** ensures that our algorithm is robust. Specifically, if our algorithm is confident that its current cluster-center estimate is close to the true cluster mean (i.e. there are many points nearby), it will confidently improve its estimate by taking a large step in the right direction (where the step size and direction are determined mainly by the nearby points). If our algorithm is not confident about being close to the cluster mean, it will tentatively improve its estimate by taking a small step in the right direction (where the step size and direction are small since the majority of points are far away and thus do not change the current estimate).

The convergence guarantee (Theorem 1) for our clustering method assumes the following:
- **Assumption 1**: The cluster-center initializations are sufficiently close to the true cluster means. *For all points $x_i$ in cluster $k$, $\mathbb{E}\|v_{k,0} - \mathbb{E}x_i\|^2 \leq \rho^2$.*
- **Assumption 2**: The variance of the points being clustered is bounded. *For all points $x_i$, $\mathbb{E}\|x_i - \mathbb{E}x_i\|^2 \leq \rho^2$.*
- **Assumption 3**: The true cluster means are sufficiently well-separated. *For all points $x_j \not\sim x_i$, $\|\mathbb{E}x_i - \mathbb{E}x_j\| > \Delta$.*

In the following theorem, we show that, after sometimes only 1 round, **Threshold-Clustering** learns a good estimate of each cluster's mean.

**Theorem 1.** *Suppose $\{x_i\}_{i\in[N]}$ is a set of points that can be partitioned into $K$ clusters such that points within each cluster are identically distributed. Suppose also that* Assumptions [1,2,3] *hold. Given cluster-center initializations $\{v_{k,0}\}_{k\in[K]}$, running* **Threshold-Clustering** *for*

$$l = \max\left(1, \max_{i\in[N]} \log(\tfrac{\rho}{\Delta} + \tfrac{1}{n_i})/\log(1 - \tfrac{n_i^2}{2N^2})\right)$$

*rounds on $\{x_i\}_{i\in[N]}$ guarantees that*

$$\mathbb{E}\|v_{k_i} - \mathbb{E}x_i\|^2 \leq \left(34C + \frac{48C}{n_i}\right)\frac{\rho^3}{\Delta} + \frac{4\rho^2}{n_i},$$

*where $k_i$ denotes client $i$'s true cluster, $n_i$ is the number of points in $k_i$, and $C$ is a constant (see Appendix A for details).*

**Remark 1** (Estimation error). *The estimation error in Theorem 1, ignoring constants, is*

$$\mathbb{E}\|v_{k_i} - \mathbb{E}x_i\|^2 \leq \mathcal{O}\left(\frac{\rho^3}{\Delta} + \frac{\rho^2}{n_i}\right). \tag{2}$$

If we knew the identity of all points within client $i$'s cluster, we would simply take their mean as the cluster-center estimate, incurring estimation error of $\rho^2/n_i$ (i.e. the sample-mean's variance). Since we don't know the identity of points within clusters, the additional factor of $\rho^3/\Delta$ in (2) is the price we pay to learn the clusters. This additional term scales with the difficulty of the clustering problem. If true clusters are well-separated and/or the variance of the points within each cluster is small (i.e. $\Delta$ is large, $\rho^2$ is small), then the clustering problem is easier and our bound is tighter. If clusters are less well-separated and/or the variance of the points within each cluster is large, accurate clustering is more difficult and our bound weakens.

The next theorem shows that the bound in Remark 1 is tight within a factor of $(\rho/\Delta)$.

**Theorem 2** (Near-optimality of **Threshold-Clustering**). *For any algorithm $\mathcal{A}$, there exists a mixture of distributions $\mathcal{D}_1 = (\mu_1, \rho^2)$ and $\mathcal{D}_2 = (\mu_2, \rho^2)$ with $\|\mu_1 - \mu_2\| \geq \Delta$ such that the estimator $\hat{\mu}_1$ produced by $\mathcal{A}$ has an error*

$$\mathbb{E}\|\hat{\mu}_1 - \mu_1\|^2 \geq \Omega\left(\frac{\rho^4}{\Delta^2} + \frac{\rho^2}{n_i}\right).$$

# 3 Personalized Learning with Threshold-Clustering

We now leverage our clustering method to develop personalized models for the clients in each cluster via stochastic optimization. We present two variants of personalized learning: one in which a central server clusters the clients (*Personalized Federated Learning* (PFL) – Algorithm 2) and one in which the clients cluster themselves (*Personalized Decentralized Learning* (PDL) – Algorithm 3). These variants are appropriate when the clients and server respectively want to minimize their computational burden.

At a high level, our learning procedures cluster clients with the same data distributions and generate a personalized model for each cluster. We assume each client $i \in [N]$ has access to an unbiased stochastic gradient

$$\mathbb{E}[g_i(x; \zeta_i)|x] = \nabla f_i(x),$$

where $f_i$ is client $i$'s loss objective. All clients in the same cluster have the same loss objective and identically distributed gradient stochasticity. Additionally we assume

- **Assumption 4**: Loss objectives are $L$-smooth.
  *For all clients $i$ and points $x, y$, $\|\nabla f_i(x) - \nabla f_i(y)\| \leq L\|x - y\|$.*

- **Assumption 5**: Variance of gradients is bounded.
  *For all clients $i$ and points $x$, $\mathbb{E}\|g_i(x; \zeta_i) - \nabla f_i(x)\| \leq \sigma^2$.*

- **Assumption 6**: Momentums of differently distributed clients are well-separated at all points.
  $$\textit{For all } i \not\sim j, \|\mathbb{E}m_{i,t} - \mathbb{E}m_{j,t}\| > \Delta,$$
  for any choice of $\alpha$ in the definition
  $$m_{i,t} = \alpha g_i(x_{i,t-1}; \zeta_i) + (1 - \alpha)m_{i,t-1}.$$

  Here, $x_{i,t}$ denotes client $i$'s model parameters at round $t$ of SGD.

**Algorithm 2** Personalized Federated Learning with Threshold-Clustering (PFL-TC)

**Input** Learning rate $\eta$; momentum parameter $\alpha$; number of clusters $K$

1: **for** round $t \in [T]$ **do**
2:     Each client $i$ computes local updates as

$$\tilde{g}_i(x_{i,t}) = \begin{cases} g_i(x_{i,t-1}) & \text{Option 1} \\ (y_{i,t}^m - x_{i,t-1})/\eta & \text{where starting from } y_{i,t}^0 = x_{i,t-1}, & \text{Option 2} \\ \quad y_t^l = y_{i,t}^{l-1} - \eta g_i(y_{i,t}^{l-1}) & \text{for } l \in \{1, \ldots, m\}. \end{cases}$$

3:     Clients compute local momentums and communicate to server

$$\{m_{i,t} = \alpha \tilde{g}_i(x_{i,t-1}) - (1-\alpha)m_{i,t-1}\}_{i \in [N]}.$$

4:     Server runs **Threshold-Clustering**($\{m_{i,t}\}_{i \in [N]}, \{v_{k_0^t}\}_{k \in [K]}, K$) and obtains estimates, $\{v_{k^t}\}_{k \in [K]}$, of the cluster means.
5:     Server updates parameters for each cluster $k^t$ and sends updated parameters to clients

$$\{x_{i,t} = x_{i,t-1} - \eta v_{k^t}\}_{i \in k^t}.$$

---

- **Assumption 7**(Optional): The cluster-center initializations are sufficiently close to the true cluster means. *For all $i \in [N], t \in [T]$, $\mathbb{E}\|v_{k_i^t,0} - \mathbb{E}m_{i,t}\|^2 \leq c\sigma^2$,*

  for some $c \geq 0$ where $k_i^t$ denotes the cluster to which $i$ is assigned in round $t$ of SGD.

Note that Assumption 7 can be easily realized by using K-means++ [2] initialization prior to running our clustering procedure.

**Clustering Momentums**: Our algorithms cluster the momentums

$$m_{i,t} = \alpha g_i(x_{i,t-1}, \zeta_i) + (1-\alpha)m_{i,t-1}$$

of the clients $i \in [N]$. Clustering momentums instead of gradients reduces clustering mistakes. To see this, say that in one particular round, a client is misclustered and thus assigned the wrong parameters. The scaling of $g$ in the momentum by a small $\alpha$ will mitigate the effect of these wrong parameters on the client's gradient update, making it more likely that the client will be clustered correctly in the next round. The clustering routines in [5, 8] both instead cluster model parameters, which has certain advantages. For instance, in order to cluster momentums, we must assume that momentums from differently distributed clients are sufficiently well separated at all points. The clustering guarantees in [5] require only that the *optima* of the loss functions of differently distributed clients be sufficiently far apart, a more reasonable assumption. However, we believe our theory can easily be adapted to cluster model parameters.

First we present our federated algorithm, in which a central server clusters the clients, along with its convergence guarantees.

**Theorem 3.** *Given* Assumptions [4-7]*, running* **PFL-TC** *with Option 1 for $T$ rounds with learning rate*

$$\eta_i = \min\left\{ \frac{1}{3L}, \left(\frac{2(f_i(x_{i,0}) - f_i^*)}{\frac{60CL\sigma^2}{n_i}T}\right)^{\frac{1}{2}}, \left(\frac{2(f_i(x_{i,0}) - f_i^*)}{\frac{21600CK^2L^{1.5}\sigma^3}{\Delta}T}\right)^{\frac{2}{5}} \right\} \tag{3}$$

*(where $\eta_i$ is the learning rate for client $i$'s true cluster, $n_i$ is the number of clients who share client $i$'s distribution, and $C$ is a constant (see Appendix A for details)) guarantees that*

$$\frac{1}{T}\sum_{t=1}^{T}\mathbb{E}\|\nabla f_i(x_{i,t-1})\|^2 \leq 2\sqrt{\frac{120CL(f_i(x_{i,0}) - f_i^*)\sigma^2}{n_iT}} + \left(\frac{21600CK^2L^{\frac{3}{2}}\sigma^3}{\Delta}\right)^{\frac{1}{5}}\left(\frac{2(f_i(x_{i,0}) - f_i^*)}{T}\right)^{\frac{4}{5}}$$

$$+ \left(\frac{21600CK^2L^{\frac{3}{2}}\sigma^3}{\Delta}\right)^{\frac{2}{5}}\left(\frac{2(f_i(x_{i,0}) - f_i^*)}{T}\right)^{\frac{3}{5}} + \frac{6L(f_i(x_{i,0} - f_i^*)}{T}. \tag{4}$$

**Algorithm 3** Personalized Decentralized Learning with Threshold-Clustering (PDL-TC)

---

**Input** Learning rate $\eta$; momentum parameter $\alpha$; number of clusters $K$

1: **for** round $t \in [T]$ **do**
2:     Each client $i$ receives all other clients' momentums, i.e. for $j \neq i$

$$\{m_{j,t} = \alpha g_j(x_{j,t-1}) - (1-\alpha)m_{j,t-1}\}_{j \in [N]\setminus i}.$$

3:     Each client $i$ runs **Threshold-Clustering**($\{m_{i,t}\}_{i \in [N]}, m_{i,t}, 1$) and obtains an estimate, $v_{k_i^t}$, of its own cluster mean.
4:     Each client computes their own parameter update

$$x_{i,t} = x_{i,t-1} - \eta v_{k_i^t}.$$

---

131 **Remark 2** (Convergence Rate). *The rate of convergence in Theorem 3, ignoring constants and*
132 *higher order terms, is*

$$\frac{1}{T}\sum_{t=1}^{T}\mathbb{E}\|\nabla f_i(x_{i,t-1})\|^2 \lesssim \mathcal{O}\Big(\sqrt{\frac{\sigma^2}{n_i T}}\Big). \tag{5}$$

133 We note a few things. The rate in (5) is the optimal rate in $T$ for stochastic gradient descent on
134 non-convex functions [1]. In (4), the leading term's dependence on $\sqrt{\sigma^2/n_i}$ is intuitive, since
135 convergence error should increase as the variance of points in the cluster increases and decrease as
136 the number of points in the cluster increases. Also, the dependence of higher-order terms in (4)
137 on the number of clusters $K$ demonstrates that as we increase the number of clusters, clustering
138 correctly may become harder (i.e. there are more opportunities to mis-cluster). Finally, the learning
139 rate $\eta_i$ in (3) is the optimal learning rate only for client $i$'s true cluster. Therefore, we can't know $\eta_i$
140 in advance, since the point of running our algorithm is to realize the true clusters. Instead, in practice
141 we just set a uniform learning rate across all clusters that works well experimentally. However, the
142 fact that, in theory, $\eta_i$ scales with $n_i$ and $\Delta$ is intuitive. As the number of points in $i$'s cluster
143 increases, and as $i$'s cluster is farther away from the other clusters, our algorithm can more easily
144 identify $i$'s cluster and thus can increase the learning rate for that cluster.

145 We now present our personalized decentralized learning method (Algorithm 3), in which clients
146 cluster themselves. While this requires more communication than the federated version ($N^2$ instead
147 of $KN$ communications per round), (a) it does not require that cluster-center initializations be close
148 to true cluster means (**Assumption 7**) since clients simply initialize their cluster-center estimates
149 with their own momentum, and (b) its convergence guarantee (Theorem 4) is independent of $K$.

150 **Theorem 4.** *Given* Assumptions [4-6]*, running* **PDL-TC** *over $T$ rounds with learning rate*

$$\eta_i = \min\left\{\frac{1}{3L}, \left(\frac{2(f_i(x_{i,0}) - f_i^*)}{\frac{48CL\sigma^2}{n_i}T}\right)^{\frac{1}{2}}, \left(\frac{2(f_i(x_{i,0}) - f_i^*)}{\frac{1968L^{1.5}\sigma^3}{\Delta}T}\right)^{\frac{2}{5}}\right\}.$$

151 *guarantees that*

$$\frac{1}{T}\sum_{t=1}^{T}\mathbb{E}\|\nabla f_i(x_{i,t-1})\|^2 \leq 2\sqrt{\frac{96CL(f_i(x_{i,0}) - f_i^*)\sigma^2}{n_i T}} + \left(\frac{1968L^{\frac{3}{2}}\sigma^3}{\Delta}\right)^{\frac{1}{5}}\left(\frac{2(f_i(x_{i,0}) - f_i^*)}{T}\right)^{\frac{4}{5}}$$

$$+ \left(\frac{1968L^{\frac{3}{2}}\sigma^3}{\Delta}\right)^{\frac{2}{5}}\left(\frac{2(f_i(x_{i,0}) - f_i^*)}{T}\right)^{\frac{3}{5}} + \frac{6L(f_i(x_{i,0} - f_i^*)}{T}.$$

152 *Therefore, the rate of convergence, ignoring constants and higher-order terms, is*

$$\frac{1}{T}\sum_{t=1}^{T}\mathbb{E}\|\nabla f_i(x_{i,t-1})\|^2 \lesssim \mathcal{O}\Big(\sqrt{\frac{\sigma^2}{n_i T}}\Big).$$

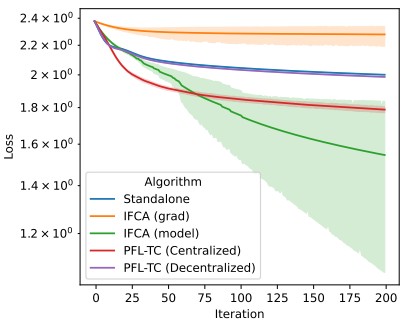

Figure 1: We report mean-squared-error loss of a model trained on the 5 listed algorithms. We run each algorithm for 5 trials.

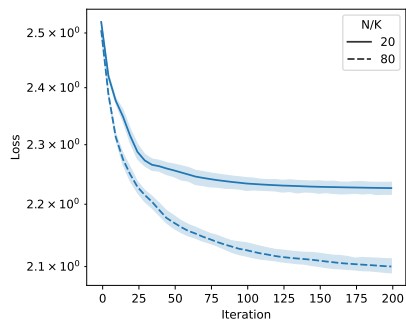

Figure 2: We report the effect of cluster size, $N/K$, on a model's accuracy when trained with PFL-TC.

## 4 Preliminary Simulations

In this section, we compare our algorithms PFL-TC (Algorithm 2) and PDL-TC (Algorithm 3) to three baselines, and display the results in Figure 1. For the first baseline (Standalone), all agents train only on their own data, and we display the average loss of their models. The other two baselines are variants of IFCA, a SOTA algorithm for cluster-based personalized federated learning proposed in [5]. In IFCA(model), agents compute their own parameter updates and the server averages them per-cluster. In IFCA(grad), the server computes per-cluster parameter updates by averaging gradients from the clients. We use the synthetic dataset described in [5] for these experiments.

Our experimental setup is the following. We set the number of clusters $K = 4$, the number of clients per cluster $n = 9$, the feature dimensionality $d = 10$, and we run all experiments for 5 trials. Hence $N = nK$. For each client $i$ in cluster $k$, we devise a linear model

$$y_i = a_i^T x_k^*,$$

where $a_i \in \mathbb{R}^{d \times n}$ is a random matrix containing agent $i$'s samples, and the per-cluster optima $x_k^* \in \mathbb{R}^d$ are generated from a binomial distribution with $\{0, 100\}$ in each coordinate. Before training we initialize the parameters for each client $\{x_{i,0}\}_{i=1}^N$ from a standard Gaussian distribution. Finally, we evaluate the output $\{x_{i,T}\}_{i=1}^N$ of all algorithms with mean-squared-error loss

$$\frac{1}{N} \sum_{i=1:N,\ i \in k,\ k \in [K]} \|x_{i,T} - x_k^\star\|_2^2.$$

In Figure 1, we see that PFL-TC and PDL-TC achieve lower loss than IFCA(grad). They also have significantly lower variance than IFCA(model) and are less sensitive to the model's initial weights.

In Figure 2, we set $d = 20$ and $n = 10$ so that any client's local data alone cannot fully determine the model weights. The results in this figure suggest that PFL-TC benefits from collaborative learning and from increasing the number of clients per cluster. Notably, this behavior is consistent with our theoretical guarantee in (4), which states that convergence error is inversely proportional to $\sqrt{N/K}$.

## 5 Conclusion

We develop two simple clustering-based algorithms to achieve personalization in federated learning. Our algorithms have optimal convergence guarantees and asymptotically match the achievable rates when the true clustering of clients is known. A current limitation of our work is that we require the momentums (effectively gradients) of differently distributed clients to be sufficiently well-separated at all points. Ideally, we would only have to impose this separation requirement on the optima of differently distributed clients' loss objectives. We are also working on more large-scale and real-world empirical benchmarking of our methods. Future directions involve formalizing the robustness properties of our clustering method and designing Byzantine-robust versions of our algorithms. Further,

our analysis can be used to show that our algorithms are incentive-compatible and lead to *stable coalitions* as in [3]. This would form a strong argument towards encouraging participants in a federated learning system. Investigating such incentives and fairness concerns is another promising future direction.

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

# Appendix

## A  Proofs

### A.1  Proof of Theorem 3

#### A.1.1  Personalized Federated Learning

We restate the main steps of the algorithm to establish notation for the proofs.

**PFL:** At round $t \in [T]$

1. Clients send momentums

$$\{m_{i,t} = \alpha g_i(x_{i,t-1}, \zeta_i) + (1-\alpha)m_{i,t-1}\}_{i=1:N}$$

to the server. The server runs **Threshold-Clustering** on $\{m_{i,t}\}_{i=1:N}$ for $M_t$ rounds, obtaining $K$ clusters $\{k^t\}$ and cluster-mean estimates $\{v_{k^t,M_t}\}_{k^t}$. The server initialized this clustering procedure with cluster-mean estimates $\{v_{k^t,0}\}_{k^t}$.

2. Server updates parameters for each cluster, $k^t$,

$$\{x_{i,t} = x_{i,t-1} - \eta v_{k^t,M_t}\}_{i\in k^t}$$

and sends them back to the clients, $i \in k^t$, in that cluster.

3. Clients update momentums

$$\{m_{i,t+1} = \alpha g_i(x_{i,t}, \zeta_i) + (1-\alpha)m_{i,t}\}_{i=1:N}.$$

**Threshold-Clustering:** At round $l \in [M_t]$

1. Server constructs $K$ balls of radii $\{\tau_{k^t,l}\}_{k^t}$ around $\{v_{k^t,l-1}\}_{k^t}$.

2. Server samples $S_{t,l}$ from $\{m_{i,t}\}_{i\in[N]}$ in per-*true*-cluster amounts $\{n_{k,t,l}\}_{k\in[K]}$ (Note: $\{k\}$ always denotes true clusters, whereas $\{k^t\}$ are the clusters obtained in round $t$). We denote $N_{t,l} = \sum_{k=1}^{K} n_{k,t,l} = |S_{t,l}|$, i.e. the total number of points sampled from $\{m_{i,t}\}_{i\in[N]}$ in thresholding round $l$; and $N_t = \sum_{l=1}^{M_t} N_{t,l}$, i.e. the total number of points sampled across all rounds of thresholding in optimization step $t$. Server sets new cluster-mean estimates $v_{k^t,l} = \frac{1}{N_{t,l}} \sum_{j=1}^{N_{t,l}} y_{j,t,l}$, where

$$y_{j,t,l} = m_{j,t}\mathbb{1}(\|v_{k^t,l-1} - m_{j,t}\| \le \tau_{k^t,l}) + v_{k^t,l-1}\mathbb{1}(\|v_{k^t,l-1} - m_{j,t}\| > \tau_{k^t,l})$$

for each $m_{j,t} \in S_{t,l}$.

**Assumptions:**

1. Server knows in advance that there are $K$ clusters.

2. $m_{i,0} = 0$ for all clients $i$, and $\alpha = 1$ and step $t = 1$ of the optimization procedure.

3. For all clients $i$ and all $x$,

$$\mathbb{E}\|g_i(x;\zeta_i) - \nabla f_i(x)\|^2 \le \sigma^2.$$

4. For all clients $i, j$ not in the same true cluster and all rounds $t \in [T]$ of the optimization cluster,
$$\|\mathbb{E}m_{i,t} - \mathbb{E}m_{j,t}\| > \Delta.$$

5. For all clients $i$, in round $t \in [T]$ of the optimization procedure,

$$\mathbb{E}\|v_{k_i^t,0} - \mathbb{E}m_{i,t}\|^2 \le \sigma^2,$$

where $k_i^t$ is the cluster to which $i$ is assigned in round $t$.

It follows from Assumption 3 above that, for all clients $i$ and rounds $t \in [T]$ of the optimization procedure,

$$\mathbb{E}\|m_{i,t} - \mathbb{E}m_{i,t}\|^2 \le \rho^2$$
$$= \alpha\sigma^2.$$

*Proof.*

$$\mathbb{E}\|m_{i,t} - \mathbb{E}m_{i,t}\|^2 = \mathbb{E}\|\alpha(g_i(x_{i,t-1}) - \nabla f_i(x_{i,t-1})) + (1-\alpha)(m_{i,t-1} - \mathbb{E}m_{i,t-1})\|^2$$
$$\le \alpha^2 \mathbb{E}\|g_i(x_{i,t-1}) - \nabla f_i(x_{i,t-1})\|^2 + (1-\alpha)^2 \mathbb{E}\|m_{i,t-1} - \mathbb{E}m_{i,t-1}\|^2$$
$$\le \alpha^2 \mathbb{E}\|g_i(x_{i,t-1}) - \nabla f_i(x_{i,t-1})\|^2 + (1-\alpha)\mathbb{E}\|m_{i,t-1} - \mathbb{E}m_{i,t-1}\|^2$$
$$\le \alpha^2\sigma^2 \sum_{q=0}^{t-1}(1-\alpha)^q$$
$$\le \alpha^2\sigma^2 \frac{(1-\alpha)^t - 1}{(1-\alpha) - 1}$$
$$\le \alpha^2\sigma^2 \frac{1}{\alpha}$$
$$= \alpha\sigma^2.$$

$\square$

## Convergence of Threshold-Clustering

**Lemma 1.** *Let $k_i^t$ be client $i$'s assigned cluster after thresholding in round $t \in [T]$ of* PFL. *Then, after*

$$l = \frac{\log\left(\frac{\rho}{\alpha\Delta} + \frac{1}{\alpha n_i}\right)}{\log\left(1 - \frac{\delta_i}{2}\right)}$$

*rounds of* Threshold-Clustering,

$$\mathbb{E}\|v_{k_i^t,l} - \mathbb{E}m_{i,t}\|^2 \le \left(467CK^2 + \frac{432CK^2}{n_{k_i}}\right)\frac{\rho^3}{\Delta} + \frac{5C\rho^2}{n_{k_i}}, \qquad (6)$$

*where $C$ is a constant.*

*Proof.* Let $k_i$ be client $i$'s true cluster.

$$\mathbb{E}\|v_{k_i^t,l} - \mathbb{E}m_{i,t}\|^2 = \mathbb{E}\left\|\frac{1}{N_{t,l}}\sum_{j\in S_{t,l}:j\sim i}(y_{j,t,l} - \mathbb{E}m_{i,t}) + \frac{1}{N_{t,l}}\sum_{j\in S_{t,l}:j\not\sim i}(y_{j,t,l} - \mathbb{E}m_{i,t})\right\|^2$$

$$\le \frac{2(1+\gamma_{k_i^t,l})}{N_{t,l}^2}\left[\left\|\sum_{j\in S_{t,l}:j\sim i}\mathbb{E}(y_{j,t,l} - m_{i,t})\right\|^2 + \mathbb{E}\left\|\sum_{j\in S_{t,l}:j\sim i}(y_{j,t,l} - \mathbb{E}y_{j,t,l})\right\|^2\right]$$

$$+ \frac{\left(1+\frac{1}{\gamma_{k_i^t,l}}\right)}{N_{t,l}^2}\mathbb{E}\left\|\sum_{j\in S_{t,l}:j\not\sim i}(y_{j,t,l} - \mathbb{E}m_{i,t})\right\|^2$$

$$\le \frac{2n_{k_i,t,l}^2(1+\gamma_{k_i^t,l})}{N_{t,l}^2}\left[\underbrace{\|\mathbb{E}_{j\sim i}(y_{j,t,l} - m_{i,t})\|^2}_{\mathcal{T}_1} + \frac{1}{n_{k_i,t,l}}\underbrace{\mathbb{E}_{j\sim i}\|y_{j,t,l} - \mathbb{E}_{j\sim i}y_{j,t,l}\|^2}_{\mathcal{T}_2}\right]$$

$$+ \frac{\left(1+\frac{1}{\gamma_{k_i^t,l}}\right)(N_{t,l} - n_{k_i,t,l})^2}{N_{t,l}^2}\underbrace{\mathbb{E}_{j\not\sim i}\|y_{j,t,l} - \mathbb{E}m_{i,t}\|^2}_{\mathcal{T}_3}. \qquad (7)$$

258    Now bound $\mathcal{T}_1, \mathcal{T}_2$, and $\mathcal{T}_3$.

260    We assume at the beginning of each thresholding round $l$ that, for all clients $i$,

$$\mathbb{E}\|v_{k_i^t, l-1} - \mathbb{E}m_{i,t}\|^2 \le c_{k_i^t, l}^2.$$

261    We also set

262    $\delta_{k,t,l} = (\frac{n_{k,t,l}}{N_{t,l}})^2$, where $k$ denotes an arbitrary *true* cluster

263    $\tau_{k_i^t, l}^2 = \frac{\sqrt{\delta_{k_i,t,l}}(c_{k_i^t,l}^2 + \rho^2)\Delta}{\rho}$, where $i$ denotes an arbitrary client

264    • and ensure that, for all $i$, $\tau_{k_i^t,l}^2 + c_{k_i^t,l}^2 + \rho^2 \le \frac{\Delta^2}{12}$ and $\tau_{k_i^t,l}^2 \le \frac{\Delta^2}{18}$.

265    Bound $\mathcal{T}_1$:

267    We will break this into two cases: 1) where clients $i \sim j$ have the same history of parame-
268    ter updates (i.e. have always been assigned to the same cluster in past rounds), and 2) where they
269    don't. Then we will note that

$$\|\mathbb{E}_{j\sim i}(y_{j,t,l} - m_{i,t})\|^2 = \|\mathbb{E}_{j\sim i}(y_{j,t,l} - m_{i,t})|\text{Case 1}\|^2\mathbb{P}(\text{Case 1}) + \|\mathbb{E}_{j\sim i}(y_{j,t,l} - m_{i,t})|\text{Case 2}\|^2\mathbb{P}(\text{Case 2}),$$

270    and evaluate each of these components.

272    Case 1: At round $t$, clients $i \sim j$ have same history of parameter updates, i.e. $x_{i,t'} = x_{j,t'}$
273    for all $t' < t$. As a reminder about notation, $k_j^t$ denotes the cluster to which $j$ is assigned after
274    thresholding in round $t \in [T]$ of the optimization procedure. In this case

$$\|\mathbb{E}_{j\sim i}(y_{j,t,l} - m_{i,t})\|^2 \le (\mathbb{E}_{j\sim i}\|y_{j,t,l} - m_{j,t}\|)^2$$

$$= \left[\mathbb{E}_{j\sim i}[\|v_{k_i^t,l-1} - m_{j,t}\|\mathbb{1}(\|v_{k_i^t,l-1} - m_{j,t}\| > \tau_{k_i^t,l})]\right]^2$$

$$\le \left[\frac{\mathbb{E}_{j\sim i}[\|v_{k_i^t,l-1} - m_{j,t}\|^2\mathbb{1}(\|v_{k_i^t,l-1} - m_{j,t}\| > \tau_{k_i^t,l})]}{\tau_{k_i^t,l}}\right]^2$$

$$\le \left[\frac{\mathbb{E}_{j\sim i}\|v_{k_i^t,l-1} - m_{j,t}\|^2}{\tau_{k_i^t,l}}\right]^2$$

$$\le \left[\frac{2\mathbb{E}_{j\sim i}\|v_{k_i^t,l-1} - \mathbb{E}_{j\sim i}m_{j,t}\|^2 + 2\mathbb{E}_{j\sim i}\|m_{j,t} - \mathbb{E}_{j\sim i}m_{j,t}\|^2}{\tau_{k_i^t,l}}\right]^2$$

$$\le \left[\frac{2\mathbb{E}\|v_{k_i^t,l-1} - \mathbb{E}m_{i,t}\|^2 + 2\mathbb{E}\|m_{j,t} - \mathbb{E}m_{j,t}\|^2}{\tau_{k_i^t,l}}\right]^2$$

$$\le \left[\frac{2c_{k_i^t,l}^2 + 2\rho^2}{\tau_{k_i^t,l}}\right]^2$$

$$= \frac{4(c_{k_i^t,l}^2 + \rho^2)^2}{\tau_{k_i^t,l}^2}.$$

275    Case 2: At round $t$, clients $i \sim j$ have different history of parameter updates, i.e. $x_{i,t} \ne x_{j,t}$ for
276    some $t' < t$. This means that at some time $t' < t$, $i$ and $j$ were first assigned to different clusters.
277    This could have happened in one of three ways: 1) $i$ was assigned to the correct cluster and $j$ to the
278    incorrect cluster, 2) $j$ was assigned to the correct cluster and $i$ to the incorrect one, 3) $i$ and $j$ were
279    both assigned to *different*, incorrect clusters.

280    Say $j$ was assigned to a specific incorrect cluster $k_{j'}^{t'}$ at time $t'$. We take am 'incorrect' here to mean
281    that $k_{j'}^{t'}$ contains at least one client $j' \nsim j$. Then

$$\|m_{j,t'} - v_{k_{j'}^{t'}, M_{t'}}\| \le \tau_{k_{j'}^{t'}, M_{t'}}.$$

282 If this happens, then

$$2\|m_{j,t'} - \mathbb{E}m_{j,t'}\|^2 \geq \|\mathbb{E}m_{j,t'} - v_{k_{j'}^{t'},M_{t'}}\|^2 - 2\|m_{j,t'} - v_{k_{j'}^{t'},M_{t'}}\|^2$$

$$\geq \|\mathbb{E}m_{j,t'} - v_{k_{j'}^{t'},M_{t'}}\|^2 - 2\tau_{k_{j'}^{t'},M_{t'}}^2$$

$$\geq \frac{1}{3}\|\mathbb{E}m_{j,t'} - \mathbb{E}m_{j',t'}\|^2 - \|\mathbb{E}m_{j',t'} - m_{j',t'}\|^2 - \|m_{j',t'} - v_{k_{j'}^{t'},M_{t'}}\|^2 - 2\tau_{k_{j'}^{t'},M_{t'}}^2$$

$$\geq \frac{1}{3}\|\mathbb{E}m_{j,t'} - \mathbb{E}m_{j',t'}\|^2 - \|\mathbb{E}m_{j',t'} - m_{j',t'}\|^2 - 3\tau_{k_{j'}^{t'},M_{t'}}^2$$

$$\geq \frac{\Delta^2}{3} - \|\mathbb{E}m_{j',t'} - m_{j',t'}\|^2 - 3\tau_{k_{j'}^{t'},M_{t'}}^2.$$

283 Equivalently

$$2\|m_{j,t'} - \mathbb{E}m_{j,t'}\|^2 + \|m_{j',t'} - \mathbb{E}m_{j',t'}\|^2 \geq \frac{\Delta^2}{3} - 3\tau_{k_{j'}^{t'},M_{t'}}^2.$$

284 The probability of this event occurring is

$$\mathbb{P}\left(2\|m_{j,t'} - \mathbb{E}m_{j,t'}\|^2 + \|m_{j',t'} - \mathbb{E}m_{j',t'}\|^2 \geq \frac{\Delta^2}{3} - 3\tau_{k_{j'}^{t'},M_{t'}}^2\right)$$

$$\leq \frac{2\mathbb{E}\|m_{j,t'} - \mathbb{E}m_{j,t'}\|^2 + \mathbb{E}\|m_{j',t'} - \mathbb{E}m_{j',t'}\|^2}{\frac{\Delta^2}{3} - 3\tau_{k_{j'}^{t'},M_{t'}}^2}$$

$$\leq \frac{3\rho^2}{\frac{\Delta^2}{3} - 3\tau_{k_{j'}^{t'},M_{t'}}^2}.$$

285 Since $i$ and $j$ can both be assigned to a maximum of $K$ different clusters,

$$\mathbb{P}(\text{Case 2}) = \mathbb{P}(i, j, \text{ or both assigned incorrectly, and differently, in round } t' < t) \leq \left(\frac{K!}{(K-2)!}\right)\frac{3\rho^2}{\frac{\Delta^2}{3} - 3\tau_{k_{j'}^{t'},M_{t'}}^2}$$

$$= \frac{3K(K-1)\rho^2}{\frac{\Delta^2}{3} - 3\tau_{k_{j'}^{t'},M_{t'}}^2}$$

$$= \frac{3K^2\rho^2}{\frac{\Delta^2}{3} - 3\tau_{k_{j'}^{t'},M_{t'}}^2}$$

$$\leq \frac{18K^2\rho^2}{\Delta^2}.$$

286 The bound on $\mathcal{T}_1$ in this case is

$$\|\mathbb{E}_{j\sim i}(y_{j,t,l} - m_{i,t})\|^2 \leq \mathbb{E}_{j\sim i}\|y_{j,t,l} - m_{i,t}\|^2$$

$$= \mathbb{E}[\|m_{j,t} - m_{i,t}\|^2 \mathbb{1}(\|m_{j,t} - v_{k_i^t,l-1}\| \leq \tau_{k_i^t,l})]$$

$$+ \mathbb{E}[\|v_{k_i^t,l-1} - m_{i,t}\|^2 \mathbb{1}(\|m_{j,t} - v_{k_i^t,l-1}\| > \tau_{k_i^t,l})]$$

$$\leq \mathbb{E}[\|m_{j,t} - m_{i,t}\|^2 \mathbb{1}(\|m_{j,t} - v_{k_i^t,l-1}\| \leq \tau_{k_i^t,l})] + \mathbb{E}\|v_{k_i^t,l-1} - m_{i,t}\|^2.$$

287 For the first part of this bound, if $\|m_{j,t} - v_{k_i^t,l-1}\| \leq \tau_{k_i^t,l}$,

$$\mathbb{E}\|m_{j,t} - m_{i,t}\|^2 \leq 2\mathbb{E}\|m_{j,t} - v_{k_i^t,l-1}\|^2 + 2\mathbb{E}\|v_{k_i^t,l-1} - m_{i,t}\|^2$$

$$\leq 2\tau_{k_i^t,l}^2 + 4\mathbb{E}\|v_{k_i^t,l-1} - \mathbb{E}m_{i,t}\|^2 + 4\mathbb{E}\|m_{i,t} - \mathbb{E}m_{i,t}\|^2$$

$$\leq 2\tau_{k_i^t,l}^2 + 4c_{k_i^t,l}^2 + 4\rho^2.$$

288 For the second part of the bound,

$$\mathbb{E}\|v_{k_i^t,l-1} - m_{i,t}\|^2 \leq 2\mathbb{E}\|v_{k_i^t,l-1} - \mathbb{E}m_{i,t}\|^2 + 2\mathbb{E}\|m_{i,t} - \mathbb{E}m_{i,t}\|^2$$

$$\leq 2c_{k_i^t,l}^2 + 2\rho^2.$$

289 Therefore

$$\|\mathbb{E}_{j\sim i}(y_{j,t,l} - m_{i,t})\|^2 \le 2\tau_{k_i^t,l}^2 + 6c_{k_i^t,l}^2 + 6\rho^2.$$

290 Combining Cases 1 and 2 to finally bound $\mathcal{T}_1$,

$$\|\mathbb{E}_{j\sim i}(y_{j,t,l} - m_{i,t})\|^2 = \|\mathbb{E}_{j\sim i}(y_{j,t,l} - m_{i,t})|\text{Case 1}\|^2 \mathbb{P}(\text{Case 1}) + \|\mathbb{E}_{j\sim i}(y_{j,t,l} - m_{i,t})|\text{Case 2}\|^2 \mathbb{P}(\text{Case 2})$$
$$\le \frac{4(c_{k_i^t,l}^2 + \rho^2)^2}{\tau_{k_i^t,l}^2} + (2\tau_{k_i^t,l}^2 + 6c_{k_i^t,l}^2 + 6\rho^2)\frac{18K^2\rho^2}{\Delta^2}$$
$$\le \frac{4(c_{k_i^t,l}^2 + \rho^2)\rho}{\sqrt{\delta_{k_i,t,l}}\Delta} + (2\tau_{k_i^t,l}^2 + 6c_{k_i^t,l}^2 + 6\rho^2)\frac{18K^2\rho^2}{\Delta^2}$$
$$\le \frac{4(c_{k_i^t,l}^2 + \rho^2)\rho}{\sqrt{\delta_{k_i,t,l}}\Delta} + \frac{36K^2\sqrt{\delta_{k_i,t,l}}(c_{k_i^t,l}^2 + \rho^2)\rho}{\Delta} + 108K^2(c_{k_i^t,l}^2 + \rho^2)\frac{\rho^2}{\Delta^2}$$
$$= \left(\frac{4(c_{k_i^t,l}^2 + \rho^2)}{\sqrt{\delta_{k_i,t,l}}} + 36K^2\sqrt{\delta_{k_i,t,l}}(c_{k_i^t,l}^2 + \rho^2)\right)\frac{\rho}{\Delta} + 108K^2(c_{k_i^t,l}^2 + \rho^2)\frac{\rho^2}{\Delta^2}.$$

291 Bound $\mathcal{T}_2$:

$$\mathbb{E}_{j\sim i}\|y_{j,t,l} - \mathbb{E}_{j\sim i}y_{j,t,l}\|^2 \le \mathbb{E}[\|m_{j,t} - \mathbb{E}m_{j,t}\|^2 \mathbb{1}(\|v_{k_i^t,l-1} - m_{j,t}\| \le \tau_{k_i^t,l})]$$
$$+ \mathbb{E}[\|v_{k_i^t,l-1} - \mathbb{E}v_{k_i^t,l-1}\|^2 \mathbb{1}(\|v_{k_i^t,l-1} - m_{j,t}\| > \tau_{k_i^t,l})]$$
$$\le \rho^2 + \mathbb{E}[(2\|v_{k_i^t,l-1} - \mathbb{E}m_{i,t}\|^2 + 4\mathbb{E}\|m_{i,t} - \mathbb{E}m_{i,t}\|^2 + 4\|v_{k_i^t,l-1} - \mathbb{E}m_{i,t}\|^2)\cdot$$
$$\mathbb{1}(\|v_{k_i^t,l-1} - m_{j,t}\| > \tau_{k_i^t,l})]$$
$$\le \rho^2 + (6c_{k_i^t,l}^2 + 4\rho^2)\mathbb{P}(\|v_{k_i^t,l-1} - m_{j,t}\| > \tau_{k_i^t,l}).$$

292 To evaluate the probability in this bound, we have to again handle cases (the same cases as for $\mathcal{T}_1$).
293
294 Case 1: At round $t$, clients $i \sim j$ have same history of parameter updates, i.e. $x_{i,t'} = x_{j,t'}$
295 for all $t' < t$. In this case,

$$\mathbb{P}(\|v_{k_i^t,l-1} - m_{j,t}\| > \tau_{k_i^t,l}|\text{Case 1}) = \mathbb{P}(\|v_{k_i^t,l-1} - m_{i,t}\| > \tau_{k_i^t,l})$$
$$\le \frac{\mathbb{E}\|v_{k_i^t,l-1} - m_{i,t}\|^2}{\tau_{k_i^t,l}^2}$$
$$\le \frac{2\mathbb{E}\|v_{k_i^t,l-1} - \mathbb{E}m_{i,t}\|^2 + 2\mathbb{E}\|m_{i,t} - \mathbb{E}m_{i,t}\|^2}{\tau_{k_i^t,l}^2}$$
$$\le \frac{2(c_{k_i^t,l}^2 + \rho^2)}{\tau_{k_i^t,l}^2}.$$

296 Case 2: At round $t$, clients $i \sim j$ have different history of parameter updates, i.e. $x_{i,t'} \ne x_{j,t'}$ for
297 some $t' < t$. The proof for Case 2 in bounding $\mathcal{T}_1$ can be applied exactly here. That is,

$$\mathbb{P}(\text{Case 2}) \le \frac{18K^2\rho^2}{\Delta^2}.$$

298 Therefore,

$$\mathbb{P}(\|v_{k_i^t,l-1} - m_{j,t}\| > \tau_{k_i^t,l})$$
$$= \mathbb{P}(\|v_{k_i^t,l-1} - m_{j,t}\| > \tau_{k_i^t,l}|\text{Case 1})\mathbb{P}(\text{Case 1}) + \mathbb{P}(\|v_{k_i^t,l-1} - m_{j,t}\| > \tau_{k_i^t,l}|\text{Case 2})\mathbb{P}(\text{Case 2})$$
$$\le \mathbb{P}(\|v_{k_i^t,l-1} - m_{j,t}\| > \tau_{k_i^t,l}|\text{Case 1}) + \mathbb{P}(\text{Case 2})$$
$$\le \frac{2c_{k_i^t,l}^2 + 2\rho^2}{\tau_{k_i^t,l}^2} + \frac{18K^2\rho^2}{\Delta^2}.$$

Incorporating this bound on the probability into a bound on $\mathcal{T}_2$,

$$\mathbb{E}_{j\sim i}\|y_{j,t,l} - \mathbb{E}_{j\sim i}y_{j,t,l}\|^2 \leq \rho^2 + (6c_{k_i^t,l}^2 + 4\rho^2)\left(\frac{2(c_{k_i^t,l}^2 + \rho^2)}{\tau_{k_i^t,l}^2} + \frac{18K^2\rho^2}{\Delta^2}\right)$$

$$\leq \rho^2 + \left(\frac{12(c_{k_i^t,l}^2 + \rho^2)}{\sqrt{\delta_{k_i,t,l}}}\right)\frac{\rho}{\Delta} + 108K^2(c_{k_i^t,l}^2 + \rho^2)\frac{\rho^2}{\Delta^2}.$$

Bound $\mathcal{T}_3$:

$$\mathbb{E}_{j\nsim i}\|y_{j,t,l} - \mathbb{E}m_{i,t}\|^2 \leq (1 + \beta_{k_i^t,l})\mathbb{E}\|v_{k_i^t,l-1} - \mathbb{E}m_{i,t}\|^2 + \left(1 + \frac{1}{\beta_{k_i^t,l}}\right)\mathbb{E}_{j\nsim i}\|y_{j,t,l} - v_{k_i^t,l-1}\|^2$$

$$\leq (1 + \beta_{k_i^t,l})(c_{k_i^t,l})^2 + \left(1 + \frac{1}{\beta_{k_i^t,l}}\right)\mathbb{E}_{j\nsim i}\|y_{j,t,l} - v_{k_i^t,l-1}\|^2$$

$$= (1 + \beta_{k_i^t,l})(c_{k_i^t,l})^2 + \left(1 + \frac{1}{\beta_{k_i^t,l}}\right)\mathbb{E}_{j\nsim i}[\|m_{j,t} - v_{k_i^t,l-1}\|^2\mathbb{1}\{\|m_{j,t} - v_{k_i^t,l-1}\| \leq \tau_{k_i^t,l}\}]$$

$$\leq (1 + \beta_{k_i^t,l})(c_{k_i^t,l})^2 + \left(1 + \frac{1}{\beta_{k_i^t,l}}\right)\tau_{k_i^t,l}^2\mathbb{P}_{j\nsim i}(\|m_{j,t} - v_{k_i^t,l-1}\| \leq \tau_{k_i^t,l}).$$

If $\|m_{j,t} - v_{k_i^t,l-1}\| \leq \tau_{k_i^t,l}$, then

$$\Delta^2 \leq \|\mathbb{E}m_{j,t} - \mathbb{E}m_{i,t}\|^2 \leq 3(\|m_{j,t} - \mathbb{E}m_{j,t}\|^2 + \|m_{j,t} - \mathbb{E}v_{k_i^t,l-1}\|^2 + \|\mathbb{E}v_{k_i^t,l-1} - \mathbb{E}m_{i,t}\|^2)$$

$$\leq 3(\|m_{j,t} - \mathbb{E}m_{j,t}\|^2 + 2\|m_{j,t} - \mathbb{E}m_{j,t}\|^2 + 2\|\mathbb{E}m_{j,t} - \mathbb{E}v_{k_i^t,l-1}\|^2$$

$$+ 2\mathbb{E}\|v_{k_i^t,l-1} - \mathbb{E}m_{i,t}\|^2 + 2\mathbb{E}\|m_{i,t} - \mathbb{E}m_{i,t}\|^2)$$

$$\leq 3(3\|m_{j,t} - \mathbb{E}m_{j,t}\|^2 + 2\tau_{k_i^t,l}^2 + 2c_{k_i^t,l}^2 + 2\rho^2).$$

The probability of this event is

$$\mathbb{P}\left(\|m_{j,t} - \mathbb{E}m_{j,t}\|^2 \geq \frac{\Delta^2}{9} - \frac{2(\tau_{k_i^t,l}^2 + c_{k_i^t,l}^2 + \rho^2)}{3}\right) \leq \frac{\mathbb{E}\|m_{j,t} - \mathbb{E}m_{j,t}\|^2}{\frac{\Delta^2}{9} - \frac{2\tau_{k_i^t,l}^2 + 2c_{k_i^t,l}^2 + 2\rho^2}{3}}$$

$$\leq \frac{\rho^2}{\frac{\Delta^2}{9} - \frac{2(\tau_{k_i^t,l}^2 + c_{k_i^t,l}^2 + \rho^2)}{3}}$$

$$\leq \frac{18\rho^2}{\Delta^2}.$$

Therefore

$$\mathbb{E}_{j\nsim i}\|y_{j,t,l} - \mathbb{E}m_{i,t}\|^2 \leq (1 + \beta_{k_i^t,l})c_{k_i^t,l}^2 + \left(1 + \frac{1}{\beta_{k_i^t,l}}\right)\tau_{k_i^t,l}^2\mathbb{P}_{j\nsim i}(\|m_{j,t} - v_{k_i^t,l-1}\| \leq \tau_{k_i^t,l})$$

$$\leq (1 + \beta_{k_i^t,l})c_{k_i^t,l}^2 + \left(1 + \frac{1}{\beta_{k_i^t,l}}\right)\frac{18\tau_{k_i^t,l}^2\rho^2}{\Delta^2}$$

$$\leq (1 + \beta_{k_i^t,l})c_{k_i^t,l}^2 + \left(1 + \frac{1}{\beta_{k_i^t,l}}\right)\frac{18\sqrt{\delta_{k_i,t,l}}(\rho^2 + c_{k_i^t,l}^2)\rho}{\Delta}.$$

 Now apply the bounds on $\mathcal{T}_1$, $\mathcal{T}_2$, and $\mathcal{T}_3$ to (7), and set $\gamma_{k_i^t,l} = \frac{1}{\sqrt{\delta_{k_i,t,l}}}$ and $\beta_{k_i^t,l} = \sqrt{\delta_{k_i,t,l}}$.

$$\mathbb{E}\|v_{k_i^t,l} - \mathbb{E}m_{i,t}\|^2$$

$$\leq \frac{2n_{k_i,t,l}^2(1+\gamma_{k_i^t,l})}{N_{t,l}^2}\left[\underbrace{\left(\frac{4(c_{k_i^t,l}^2+\rho^2)}{\sqrt{\delta_{k_i,t,l}}} + 36K^2\sqrt{\delta_{k_i,t,l}}(c_{k_i^t,l}^2+\rho^2)\right)\frac{\rho}{\Delta} + 108K^2(c_{k_i^t,l}^2+\rho^2)\frac{\rho^2}{\Delta^2}}_{\mathcal{T}_1}\right.$$

$$\left. + \frac{1}{n_{k_i,t,l}}\underbrace{\left(\rho^2 + \left(\frac{12(c_{k_i^t,l}^2+\rho^2)}{\sqrt{\delta_{k_i,t,l}}}\right)\frac{\rho}{\Delta} + 108K^2(c_{k_i^t,l}^2+\rho^2)\frac{\rho^2}{\Delta^2}\right)}_{\mathcal{T}_2}\right]$$

$$+ \frac{\left(1+\frac{1}{\gamma_{k_i^t,l}}\right)(N_{t,l}-n_{k_i,t,l})^2}{N_{t,l}^2}\left[\underbrace{(1+\beta_{k_i^t,l})c_{k_i^t,l}^2 + \left(1+\frac{1}{\beta_{k_i^t,l}}\right)\frac{18\sqrt{\delta_{k_i,t,l}}(\rho^2+c_{k_i^t,l}^2)\rho}{\Delta}}_{\mathcal{T}_3}\right]$$

$$\leq 2\delta_{k_i,t,l}\left(1+\frac{1}{\sqrt{\delta_{k_i,t,l}}}\right)\left[\underbrace{\left(\frac{4(c_{k_i^t,l}^2+\rho^2)}{\sqrt{\delta_{k_i,t,l}}} + 36K^2\sqrt{\delta_{k_i,t,l}}(c_{k_i^t,l}^2+\rho^2)\right)\frac{\rho}{\Delta} + 108K^2(c_{k_i^t,l}^2+\rho^2)\frac{\rho^2}{\Delta^2}}_{\mathcal{T}_1}\right.$$

$$\left. + \frac{1}{n_{k_i,t,l}}\underbrace{\left(\rho^2 + (4c_{k_i^t,l}^2+2\rho^2)\left(\frac{\rho}{\sqrt{\delta_{k_i,t,l}}\Delta} + \frac{108K\rho^2}{\Delta^2}\right)\right)}_{\mathcal{T}_2}\right]$$

$$+ (1+\sqrt{\delta_{k_i,t,l}})(1-\sqrt{\delta_{k_i,t,l}})^2\left[\underbrace{(1+\sqrt{\delta_{k_i,t,l}})c_{k_i^t,l}^2 + \left(1+\frac{1}{\sqrt{\delta_{k_i,t,l}}}\right)\frac{18\sqrt{\delta_{k_i,t,l}}(\rho^2+c_{k_i^t,l}^2)\rho}{\Delta}}_{\mathcal{T}_3}\right]$$

$$\leq 2(\delta_{k_i,t,l}+\sqrt{\delta_{k_i,t,l}})\left[\underbrace{\left(\frac{4(c_{k_i^t,l}^2+\rho^2)}{\sqrt{\delta_{k_i,t,l}}} + 36K^2\sqrt{\delta_{k_i,t,l}}(c_{k_i^t,l}^2+\rho^2)\right)\frac{\rho}{\Delta} + 108K^2(c_{k_i^t,l}^2+\rho^2)\frac{\rho^2}{\Delta^2}}_{\mathcal{T}_1}\right.$$

$$\left. + \frac{1}{n_{k_i,t,l}}\underbrace{\left(\rho^2 + \left(\frac{12(c_{k_i^t,l}^2+\rho^2)}{\sqrt{\delta_{k_i,t,l}}}\right)\frac{\rho}{\Delta} + 108K^2(c_{k_i^t,l}^2+\rho^2)\frac{\rho^2}{\Delta^2}\right)}_{\mathcal{T}_2}\right] + (1-\delta_{k_i,t,l})\left[c_{k_i^t,l}^2 + \frac{18(c_{k_i^t,l}^2+\rho^2)\rho}{\Delta}\right]$$

$$\leq \left[\left(2(\delta_{k_i,t,l}+\sqrt{\delta_{k_i,t,l}})\left(\frac{4}{\sqrt{\delta_{k_i,t,l}}} + 36K^2\sqrt{\delta_{k_i,t,l}} + \frac{12}{\sqrt{\delta_{k_i,t,l}}n_{k_i,t,l}}\right) + 18(1-\delta_{k_i,t,l})\right)\frac{\rho}{\Delta}\right.$$

$$+ \left(2(\delta_{k_i,t,l}+\sqrt{\delta_{k_i,t,l}})\left(108K^2 + \frac{108K^2}{n_{k_i,t,l}}\right)\right)\frac{\rho^2}{\Delta^2} + (1-\delta_{k_i,t,l})\bigg]c_{k_i^t,l}^2$$

$$+ \left[\left(2(\delta_{k_i,t,l}+\sqrt{\delta_{k_i,t,l}})\left(\frac{4}{\sqrt{\delta_{k_i,t,l}}} + 36K^2\sqrt{\delta_{k_i,t,l}} + \frac{12}{\sqrt{\delta_{k_i,t,l}}n_{k_i,t,l}}\right) + 18(1-\delta_{k_i,t,l})\right)\frac{\rho}{\Delta}\right.$$

$$+ \left(2(\delta_{k_i,t,l}+\sqrt{\delta_{k_i,t,l}})\left(108K^2 + \frac{108K^2}{n_{k_i,t,l}}\right)\right)\frac{\rho^2}{\Delta^2} + \frac{2(\delta_{k_i,t,l}+\sqrt{\delta_{k_i,t,l}})}{n_{k_i,t,l}}\bigg]\rho^2$$

$$\leq \left[\left(34 + 144K^2 + \frac{48}{n_{k_i,t,l}}\right)\frac{\rho}{\Delta} + \left(432K^2 + \frac{432K^2}{n_{k_i,t,l}}\right)\frac{\rho^2}{\Delta^2} + (1-\delta_{k_i,t,l})\right]c_{k_i^t,l}^2$$

$$+ \left[\left(34 + 144K^2 + \frac{48}{n_{k_i,t,l}}\right)\frac{\rho}{\Delta} + \left(432K^2 + \frac{432K^2}{n_{k_i,t,l}}\right)\frac{\rho^2}{\Delta^2} + \frac{4}{n_{k_i,t,l}}\right]\rho^2.$$

Now set $\Delta$ such that the $\frac{\rho}{\Delta}$ and $\frac{\rho^2}{\Delta^2}$ coefficients of $c_{k_i^t,l}^2$ are bounded above by $\frac{\delta_{k_i,t,l}}{4}$. This way, the entire coefficient of $c_{k_i^t,l}^2$ will be bounded above by $1 - \frac{\delta_{k_i,t,l}}{2}$.

$$\Delta > \max_{i \in [N], t \in [T], l \in [M_t]} \max \left\{ \frac{4 \left( 34 + 144K^2 + \frac{48}{n_{k_i,t,l}} \right) \rho}{\delta_{k_i,t,l}}, \sqrt{\frac{4 \left( 432K^2 + \frac{432K^2}{n_{k_i,t,l}} \right) \rho^2}{\delta_{k_i,t,l}}} \right\}.$$

Then,

$$\mathbb{E}\|v_{k_i^t,l} - \mathbb{E}m_{i,t}\|^2 \le \left( 1 - \frac{\delta_{k_i,t,l}}{2} \right) c_{k_i^t,l}^2 + \left( \left( 34 + 144K^2 + \frac{48}{n_{k_i,t,l}} \right) \frac{\rho}{\Delta} + \left( 432K^2 + \frac{432K^2}{n_{k_i,t,l}} \right) \frac{\rho^2}{\Delta^2} + \frac{4}{n_{k_i,t,l}} \right) \rho^2.$$

$$(8)$$

Set $c_{k_i^t,l+1}^2$ to the right side of (8). Unrolling the recursion over $l$ rounds,

$$\mathbb{E}\|v_{k_i^t,l} - \mathbb{E}m_{i,t}\|^2 \le \left( 1 - \frac{\delta_{k_i,t,l}}{2} \right)^l \sigma^2$$

$$+ \left[ \left( \left( 34 + 144K^2 + \frac{48}{n_{k_i,t,l}} \right) \frac{\rho}{\Delta} + \left( 432K^2 + \frac{432K^2}{n_{k_i,t,l}} \right) \frac{\rho^2}{\Delta^2} + \frac{4}{n_{k_i,t,l}} \right) \rho^2 \right] \sum_{q=0}^{l-1} \left( 1 - \frac{\delta_{k_i,t,l}}{2} \right)^q$$

$$\le \left( 1 - \frac{\delta_{k_i,t,l}}{2} \right)^l \sigma^2$$

$$+ \min_{l \in [M_t]} \left[ \left( \left( 34 + 144K^2 + \frac{48}{n_{k_i,t,l}} \right) \frac{\rho}{\Delta} + \left( 432K^2 + \frac{432K^2}{n_{k_i,t,l}} \right) \frac{\rho^2}{\Delta^2} + \frac{4}{n_{k_i,t,l}} \right) \rho^2 \right]$$

$$\cdot \sum_{q=0}^{l-1} \left( 1 - \frac{\delta_{k_i,t,l}}{2} \right)^q.$$

Let $n_i$ be the number of clients who share client $i$'s distribution, and let $\delta_i = \frac{n_i}{N}$ Then

$$\left( 1 - \frac{\delta_{k_i,t,l}}{2} \right)^l \sigma^2 + \min_{l \in [M_t]} \left[ \left( \left( 34 + 144K^2 + \frac{48}{n_{k_i,t,l}} \right) \frac{\rho}{\Delta} + \left( 432K^2 + \frac{432K^2}{n_{k_i,t,l}} \right) \frac{\rho^2}{\Delta^2} + \frac{4}{n_{k_i,t,l}} \right) \rho^2 \right] \sum_{q=0}^{l-1} \left( 1 - \frac{\delta_{k_i,t,l}}{2} \right)^q$$

$$\le C \left[ \left( 1 - \frac{\delta_i}{2} \right)^l \sigma^2 + \left[ \left( \left( 34 + 144K^2 + \frac{48}{n_i} \right) \frac{\rho}{\Delta} + \left( 432K^2 + \frac{432K^2}{n_i} \right) \frac{\rho^2}{\Delta^2} + \frac{4}{n_i} \right) \rho^2 \right] \right],$$

where $C$ is a constant to reflect that the series above converges, $\delta_{k_i,t,l}$ is within a constant factor of $\delta_i$, and $n_{k_i,t,l}$ is within a constant factor of $n_i$. Then, running this thresholding procedure for

$$l = \frac{\log \left( \frac{\rho}{\alpha\Delta} + \frac{1}{\alpha n_i} \right)}{\log \left( 1 - \frac{\delta_i}{2} \right)}$$

rounds guarantees that

$$\mathbb{E}\|v_{k_i^t,l} - \mathbb{E}m_{i,t}\|^2 \le \left( 467CK^2 + \frac{432CK^2}{n_i} \right) \frac{\rho^3}{\Delta} + \frac{5C\rho^2}{n_i}. \tag{9}$$

$\square$

**Convergence of PFL**

*Proof.* Define $B_i$ to be the RHS of (9), and assume the learning rate $\eta \leq \frac{1}{L}$. For client $i$, by

$L$-smoothness of $f_i$,

$$f_i(x_{i,t}) \leq f_i(x_{i,t-1}) + \langle \nabla f_i(x_{i,t-1}), x_{i,t} - x_{i,t-1} \rangle + \frac{L}{2}\|x_{i,t} - x_{i,t-1}\|^2$$

$$= f_i(x_{i,t-1}) - \eta\langle \nabla f_i(x_{i,t-1}), v_{k_i^t, M_t}\rangle + \frac{L\eta^2}{2}\|v_{k_i^t, M_t}\|^2$$

$$= f_i(x_{i,t-1}) + \frac{\eta}{2}\|v_{k_i^t, M_t} - \nabla f_i(x_{i,t-1})\|^2 - \frac{\eta}{2}\|\nabla f_i(x_{i,t-1})\|^2 - \frac{\eta}{2}(1 - L\eta)\|v_{k_i^t, M_t}\|^2$$

$$\leq f_i(x_{i,t-1}) + \eta\|v_{k_i^t, M_t} - \mathbb{E}m_{i,t}\|^2 + \eta\|\mathbb{E}m_{i,t} - \nabla f_i(x_{i,t-1})\|^2 - \frac{\eta}{2}\|\nabla f_i(x_{i,t-1})\|^2 - \frac{\eta}{2}(1 - L\eta)\|v_{k_i^t, M_t}\|^2.$$
$$(10)$$

Define $\phi_{i,t} = \mathbb{E}m_{i,t} - \nabla f_i(x_{i,t-1})$. Setting $\alpha \geq 3L\eta$,

$$\|\phi_{i,t}\|^2 = \|\mathbb{E}m_{i,t} - \nabla f_i(x_{i,t-1})\|^2$$

$$= \|\alpha\mathbb{E}g_i(x_{i,t-1}) + (1-\alpha)\mathbb{E}m_{i,t-1} - \nabla f_i(x_{i,t-1})\|^2$$

$$= (1-\alpha)^2\|\mathbb{E}m_{i,t-1} - \nabla f_i(x_{i,t-1})\|^2$$

$$\leq (1-\alpha)^2(1+\alpha)\|\mathbb{E}m_{i,t-1} - \nabla f_i(x_{i,t-2})\|^2 + (1-\alpha)^2\left(1 + \frac{1}{\alpha}\right)\|\nabla f_i(x_{i,t-1}) - \nabla f_i(x_{i,t-2})\|^2$$

$$\leq (1-\alpha)^2(1+\alpha)\|\mathbb{E}m_{i,t-1} - \nabla f_i(x_{i,t-2})\|^2 + L^2(1-\alpha)\left(1 + \frac{1}{\alpha}\right)\|x_{i,t-1} - x_{i,t-2}\|^2$$

$$\leq (1-\alpha)\|\mathbb{E}m_{i,t-1} - \nabla f_i(x_{i,t-2})\|^2 + L^2\eta^2(1-\alpha)\left(1 + \frac{1}{\alpha}\right)\|v_{k_i^{t-1}, M_{t-1}}\|^2$$

$$\leq (1-\alpha)\|\phi_{i,t-1}\|^2 + \frac{L\eta}{2}(1 - L\eta)\|v_{k_i^{t-1}, M_{t-1}}\|^2. \tag{11}$$

Scale (10) by $L$, take its expectation, and add it to the expectation of (11). Then by Lemma 1,

$$\underbrace{L\mathbb{E}(f_i(x_{i,t}) - f_i(x_i^*)) + \frac{L\eta}{2}(1 - L\eta)\mathbb{E}\|v_{k_i^t, M_t}\|^2 + (1 - L\eta)\|\phi_{i,t}\|^2}_{\xi_{i,t}} + \frac{L\eta}{2}\mathbb{E}\|\nabla f_i(x_{i,t-1})\|^2$$

$$\leq L\mathbb{E}(f_i(x_{i,t-1}) - f_i(x_i^*)) + \frac{L\eta}{2}(1 - L\eta)\mathbb{E}\|v_{k_i^{t-1}, M_{t-1}}\|^2 + (1 - \alpha)\|\phi_{i,t-1}\|^2 + L\eta B_i$$

$$\leq \underbrace{L\mathbb{E}(f_i(x_{i,t-1}) - f_i(x_i^*)) + \frac{L\eta}{2}(1 - L\eta)\mathbb{E}\|v_{k_i^{t-1}, M_{t-1}}\|^2 + (1 - L\eta)\|\phi_{i,t-1}\|^2}_{\xi_{i,t-1}} + L\eta B_i. \tag{12}$$

Summing (12) over $t$ and observing that $\|\phi_{i,1}\| = \|\mathbb{E}m_{i,1} - \nabla f_i(x_{i,0})\| = 0$,

$$\frac{1}{T}\sum_{t=1}^{T} \frac{L\eta}{2}\mathbb{E}\|\nabla f_i(x_{i,t-1})\|^2 = \frac{1}{T}\left(\sum_{t=2}^{T} \frac{L\eta}{2}\mathbb{E}\|\nabla f_i(x_{i,t-1})\|^2 + \frac{L\eta}{2}\mathbb{E}\|\nabla f_i(x_{i,0})\|^2\right)$$

$$\leq \frac{1}{T}\left(\sum_{t=2}^{T}(\xi_{i,t-1} - \xi_{i,t}) + L\eta B_i\right.$$

$$+ L\mathbb{E}((f_i(x_{i,0}) - f_i(x^*)) - (f_i(x_{i,1}) - f_i(x^*)))$$

$$\left. + L\eta\|\mathbb{E}m_{i,1} - \nabla f_i(x_{i,0})\|^2 + L\eta B_i - \frac{L\eta}{2}(1 - L\eta)\mathbb{E}\|v_{k_i^t}, M_t\|^2\right)$$

$$\leq \frac{L(f_i(x_{i,0}) - f_i(x^*))}{T} + 2L\eta B_i.$$

320 Equivalently, setting $\alpha = 3L\eta$

$$\frac{1}{T}\sum_{t=1}^{T}\mathbb{E}\|\nabla f_i(x_{i,t-1})\|^2 \le \frac{2(f_i(x_{i,0}) - f_i^*)}{\eta T} + 4B_i$$

$$\le \frac{2(f_i(x_{i,0}) - f_i^*)}{\eta T} + 4\left(467CK^2 + \frac{432CK^2}{n_i}\right)\frac{\rho^3}{\Delta} + \frac{20C\rho^2}{n_i}$$

$$= \frac{2(f_i(x_{i,0}) - f_i^*)}{\eta T} + 4\left(467CK^2 + \frac{432CK^2}{n_i}\right)\frac{\alpha^{\frac{3}{2}}\sigma^3}{\Delta} + \frac{20C\alpha\sigma^2}{n_i}$$

$$\le \frac{2(f_i(x_{i,0}) - f_i^*)}{\eta T} + 4\left(467CK^2 + \frac{432CK^2}{n_i}\right)\frac{(3L\eta)^{\frac{3}{2}}\sigma^3}{\Delta} + \frac{60CL\eta\sigma^2}{n_i}$$

$$\le \frac{2(f_i(x_{i,0}) - f_i^*)}{\eta T} + 4\left(467CK^2 + \frac{432CK^2}{n_i}\right)\frac{(3L\eta)^{\frac{3}{2}}\sigma^3}{\Delta} + \frac{60CL\eta\sigma^2}{n_i}$$

$$\le \frac{2(f_i(x_{i,0}) - f_i^*)}{\eta T} + 21600CK^2\frac{(L\eta)^{\frac{3}{2}}\sigma^3}{\Delta} + \frac{60CL\eta\sigma^2}{n_i}$$

321 Choose

$$\eta = \min\left\{\frac{1}{3L}, \left(\frac{2(f_i(x_{i,0}) - f_i^*)}{\frac{60CL\sigma^2}{n_i}T}\right)^{\frac{1}{2}}, \left(\frac{2(f_i(x_{i,0}) - f_i^*)}{\frac{21600CK^2L^{1.5}\sigma^3}{\Delta}T}\right)^{\frac{2}{5}}\right\}.$$

322 Then

$$\frac{1}{T}\sum_{t=1}^{T}\mathbb{E}\|\nabla f_i(x_{i,t-1})\|^2 \le 2\sqrt{\frac{120CL(f_i(x_{i,0}) - f_i^*)\sigma^2}{n_iT}}$$

$$+ \left(\frac{21600CK^2L^{\frac{3}{2}}\sigma^3}{\Delta}\right)^{\frac{1}{5}}\left(\frac{2(f_i(x_{i,0}) - f_i^*)}{T}\right)^{\frac{4}{5}}$$

$$+ \left(\frac{21600CK^2L^{\frac{3}{2}}\sigma^3}{\Delta}\right)^{\frac{2}{5}}\left(\frac{2(f_i(x_{i,0}) - f_i^*)}{T}\right)^{\frac{3}{5}}$$

$$+ \frac{6L(f_i(x_{i,0} - f_i^*)}{T}.$$

323 $\qquad\qquad\qquad\qquad\qquad\qquad\qquad\qquad\qquad\qquad\qquad\qquad\qquad\qquad\qquad\qquad\square$

### 324 A.1.2 Personalized Decentralized Learning

325 We restate the problem setup and outline the main steps of the algorithm to establish notation for
326 the proofs.

328 **Problem Setup:** There are $N$ clients $i \in [N]$. Each client wants to optimize an $L$-smooth
329 objective $f_i$ and has access to an unbiased stochastic gradient such that, for all $x$,

$$\mathbb{E}_{\zeta_i}[g_i(x; \zeta_i)|x] = \nabla f_i(x).$$

330 All clients in the same true cluster have the same objective $f_i$, and for all clients $i$ in the same true
331 cluster $k$, $\zeta_i \sim \mathcal{P}_k$, for distributions $\{\mathcal{P}_k\}_{k\in[K]}$.

333 **PDL:** At round $t \in [T]$

334     1. Clients send momentums

$$\{m_{i,t} = \alpha g_i(x_{i,t-1}, \zeta_i) + (1 - \alpha)m_{i,t-1}\}_{i=1:N}$$

335     to each other. Each client runs thresholding on $\{m_{i,t}\}_{i=1:N}$ only around their own momen-
336     tum $m_{i,t}$ for $M_{i,t}$ rounds. That is, client $i$ sets $v_{i,t,0} = m_{i,t}$ and only maintains this cluster.
337     After thresholding, client $i$'s new cluster-mean estimate is $v_{i,t,M_t}$.

338    2. Each client $i$ updates their parameters

$$x_{i,t} = x_{i,t-1} - \eta v_{i,t,M_t}$$

339     and their momentum

$$m_{i,t+1} = \alpha g_i(x_{i,t}, \zeta_i) + (1-\alpha)m_{i,t}.$$

340 **Threshold-Clustering:** At round $l \in [M_{i,t}]$

341    1. Client $i$ samples $S_{i,t,l}$ from $\{m_{i,t}\}_{i\in[N]}$. We let $N_{i,t,l} = |S_{i,t,l}|$ and $N_{i,t} = \sum_{l=1}^{M_{i,t}} N_{i,t,l}$.
342     Additionally, let $n_{i,t,l}$ denote the number of clients from $i$'s distribution in the sample $S_{i,t,l}$.

343    2. client $i$ sets the new estimate of their own cluster center $v_{i,t,l} = \frac{1}{N_{i,t,l}} \sum_{j=1}^{N_{i,t,l}} y_{i,j,t,l}$, where

$$y_{i,j,t,l} = m_{j,t} \mathbb{1}(\|v_{i,t,l-1} - m_{j,t}\| \le \tau_{i,t,l}) + v_{i,t,l-1}\mathbb{1}(\|v_{i,t,l-1} - m_{j,t}\| > \tau_{i,t,l})$$

344     for each $m_{j,t} \in S_{i,t,l}$.

345 **Assumptions:**

346    1. For all clients $i$,

$$\mathbb{E}\|g_i(x;\zeta_i) - \nabla f_i(x)\|^2 \le \sigma^2.$$

347    2. For all clients $i, j$ not in the same true cluster and all rounds $t \in [T]$ of the optimization
348     cluster,

$$\|\mathbb{E}m_{i,t} - \mathbb{E}m_{j,t}\| > \Delta.$$

349    3. For all clients $i$ and rounds $t \in [T]$ of the optimization procedure,

$$\mathbb{E}\|m_{i,t} - \mathbb{E}m_{i,t}\|^2 \le \rho^2$$
$$= \alpha\sigma^2.$$

350 **Convergence of Thresholding Procedure**

351 **Lemma 2.** *For all rounds $l$ of* Threshold-Clustering,

$$\mathbb{E}\|v_{i,t,l} - \mathbb{E}m_{i,t}\|^2 \le C\left(\left(34 + \frac{48}{n_i}\right)\frac{\rho}{\Delta} + \frac{4}{n_i}\right)\rho^2,$$

352 *where $C$ is a constant.*

353 *Proof.* Let $k_i$ denote client $i$'s true cluster and let $n_{i,t,l} = |\{j : m_{j,t} \sim \mathcal{P}_{m_{i,t}}\}|$.

$$\mathbb{E}\|v_{i,t,l} - \mathbb{E}m_{i,t}\|^2 = \mathbb{E}\left\|\frac{1}{N_{i,t,l}} \sum_{j:m_{j,t}\sim\mathcal{P}_{m_{i,t}}} (y_{i,j,t,l} - \mathbb{E}m_{i,t}) + \frac{1}{N_{i,t,l}} \sum_{j:j\not\sim\mathcal{P}_{m_{i,t}}} (y_{i,j,t,l} - \mathbb{E}m_{i,t})\right\|^2$$

$$\le \frac{2(1+\gamma_{i,t,l})}{N_{i,t,l}^2}\left[\left\|\sum_{j:m_{j,t}\sim\mathcal{P}_{m_{i,t}}} \mathbb{E}(y_{i,j,t,l} - m_{i,t})\right\|^2 + \mathbb{E}\left\|\sum_{j:m_{j,t}\sim\mathcal{P}_{m_{i,t}}} (y_{i,j,t,l} - \mathbb{E}y_{i,j,t,l})\right\|^2\right]$$

$$+ \frac{\left(1+\frac{1}{\gamma_{i,t,l}}\right)}{N_{i,t,l}^2}\mathbb{E}\left\|\sum_{j:m_{j,t}\not\sim\mathcal{P}_{m_{i,t}}} (y_{i,j,t,l} - \mathbb{E}m_{i,t})\right\|^2$$

$$\le \frac{2n_{i,t,l}^2(1+\gamma_{i,t,l})}{N_{i,t,l}^2}\Bigg[\underbrace{\|\mathbb{E}_{j:m_{j,t}\sim\mathcal{P}_{m_{i,t}}}(y_{i,j,t,l} - m_{i,t})\|^2}_{\mathcal{T}_1}$$

$$+ \frac{1}{n_{i,t,l}}\underbrace{\mathbb{E}_{j:m_{j,t}\sim\mathcal{P}_{m_{i,t}}}\|y_{i,j,t,l} - \mathbb{E}_{j:m_{j,t}\sim\mathcal{P}_{m_{i,t}}}y_{i,j,t,l}\|^2}_{\mathcal{T}_2}\Bigg]$$

$$+ \frac{\left(1+\frac{1}{\gamma_{i,t,l}}\right)(N_{i,t,l} - n_{i,t,l})^2}{N_{i,t,l}^2}\underbrace{\mathbb{E}_{j:m_{j,t}\not\sim\mathcal{P}_{m_{i,t}}}\|y_{i,j,t,l} - \mathbb{E}m_{i,t}\|^2}_{\mathcal{T}_3}. \quad (13)$$

354 Now bound $\mathcal{T}_1, \mathcal{T}_2$, and $\mathcal{T}_3$.

355

356 We assume at the beginning of each thresholding round $l$ that, for all clients $i$,

$$\mathbb{E}\|v_{i,t,l-1} - \mathbb{E}m_{i,t}\|^2 \leq c_{i,t,l}^2.$$

357 We also set

358 $\quad$ • $\delta_{i,t,l} = (\frac{n_{i,t,l}}{N_{i,t,l}})^2$

359 $\quad$ • $\tau_{i,t,l}^2 = \frac{\sqrt{\delta_{i,t,l}}(c_{i,t,l}^2 + \rho^2)\Delta}{\rho}$

360 $\quad$ • and ensure that, for all $i$, $\tau_{i,t,l}^2 + c_{i,t,l}^2 + \rho^2 \leq \frac{\Delta^2}{12}$.

361 Bound $\mathcal{T}_1$:

$$
\begin{aligned}
\|\mathbb{E}_{j:m_{j,t}\sim\mathcal{P}_{m_{i,t}}}(y_{i,j,t,l} - m_{i,t})\|^2 &= (\mathbb{E}_{j:m_{j,t}\sim\mathcal{P}_{m_{i,t}}}\|y_{i,j,t,l} - m_{i,t}\|)^2 \\
&= \left[\mathbb{E}_{j:m_{j,t}\sim\mathcal{P}_{m_{i,t}}}\|y_{i,j,t,l} - m_{i,t}\|\right]^2 \\
&\leq \left[\mathbb{E}[\|v_{i,t,l-1} - m_{i,t}\|\mathbb{1}\{\|v_{i,t,l-1} - m_{i,t}\| > \tau_{i,t,l}\}]\right]^2 \\
&\leq \left[\frac{\mathbb{E}[\|v_{i,t,l-1} - m_{i,t}\|^2\mathbb{1}\{\|v_{i,t,l-1} - m_{i,t}\| > \tau_{i,t,l}\}]}{\tau_{i,t,l}}\right]^2 \\
&\leq \left[\frac{\mathbb{E}\|v_{i,t,l-1} - m_{i,t}\|^2}{\tau_{i,t,l}}\right]^2 \\
&= \left[\frac{2\mathbb{E}\|v_{i,t,l-1} - \mathbb{E}m_{i,t}\|^2 + 2\mathbb{E}\|m_{i,t} - \mathbb{E}m_{i,t}\|^2}{\tau_{i,t,l}}\right]^2 \\
&\leq \left[\frac{2c_{i,t,l}^2 + 2\rho^2}{\tau_{i,t,l}}\right]^2 \\
&\leq \frac{4(c_{i,t,l}^2 + \rho^2)^2}{\tau_{i,t,l}^2} \\
&\leq \frac{4(c_{i,t,l}^2 + \rho^2)\rho}{\sqrt{\delta_{i,t,l}}\Delta}.
\end{aligned}
$$

362 Bound $\mathcal{T}_2$:

$$
\begin{aligned}
\mathbb{E}_{j:m_{j,t}\sim\mathcal{P}_{m_{i,t}}}\|y_{i,j,t,l} - \mathbb{E}_{j:m_{j,t}\sim\mathcal{P}_{m_{i,t}}}y_{i,j,t,l}\|^2 &\leq \mathbb{E}\|m_{i,t} - \mathbb{E}m_{i,t}\|^2\mathbb{P}(\|v_{i,t,l-1} - m_{i,t}\| \leq \tau_{i,t,l}) \\
&\quad + \mathbb{E}\|v_{i,t,l-1} - \mathbb{E}v_{i,t,l-1}\|^2\mathbb{P}(\|v_{i,t,l-1} - m_{i,t}\| > \tau_{i,t,l}) \\
&\leq \rho^2 + 2(\mathbb{E}\|v_{i,t,l-1} - \mathbb{E}m_{i,t}\|^2 + \|\mathbb{E}m_{i,t} - \mathbb{E}v_{i,t,l-1}\|^2)\cdot \\
&\quad \mathbb{P}(\|v_{i,t,l-1} - m_{i,t}\| > \tau_{i,t,l}) \\
&\leq \rho^2 + 2(\mathbb{E}\|v_{i,t,l-1} - \mathbb{E}m_{i,t}\|^2 + 2\mathbb{E}\|m_{i,t} - \mathbb{E}m_{i,t}\|^2 \\
&\quad + 2\mathbb{E}\|v_{i,t,l-1} - \mathbb{E}m_{i,t}\|^2)\mathbb{P}(\|v_{i,t,l-1} - m_{i,t}\| > \tau_{i,t,l}) \\
&\leq \rho^2 + (6c_{i,t,l}^2 + 4\rho^2)\mathbb{P}(\|v_{i,t,l-1} - m_{i,t}\| > \tau_{i,t,l}) \\
&\leq \rho^2 + (6c_{i,t,l}^2 + 4\rho^2)\frac{2\mathbb{E}\|v_{i,t,l-1} - \mathbb{E}m_{i,t}\|^2 + 2\mathbb{E}\|m_{i,t} - \mathbb{E}m_{i,t}\|^2}{\tau_{i,t,l}^2} \\
&\leq \rho^2 + \frac{12(c_{i,t,l}^2 + \rho^2)^2}{\tau_{i,t,l}^2} \\
&\leq \rho^2 + \frac{12(c_{i,t,l}^2 + \rho^2)\rho}{\sqrt{\delta_{i,t,l}}\Delta}.
\end{aligned}
$$

363   Bound $\mathcal{T}_3$:

$$
\begin{aligned}
\mathbb{E}_{j:m_{j,t}\nsim\mathcal{P}_{m_{i,t}}}\|y_{i,j,t,l} - \mathbb{E}m_{i,t}\|^2 &\leq (1+\beta_{i,t,l})\mathbb{E}\|v_{i,t,l-1} - \mathbb{E}m_{i,t}\|^2 + \left(1 + \frac{1}{\beta_{i,t,l}}\right)\mathbb{E}_{j:m_{j,t}\nsim\mathcal{P}_{m_{i,t}}}\|y_{i,j,t,l} - v_{i,t,l-1}\|^2 \\
&\leq (1+\beta_{i,t,l})c_{i,t,l}^2 + \left(1 + \frac{1}{\beta_{i,t,l}}\right)\mathbb{E}_{j:m_{j,t}\nsim\mathcal{P}_{m_{i,t}}}\|y_{i,j,t,l} - v_{i,t,l-1}\|^2 \\
&= (1+\beta_{i,t,l})c_{i,t,l}^2 \\
&\quad + \left(1 + \frac{1}{\beta_{i,t,l}}\right)\mathbb{E}_{j:m_{j,t}\nsim\mathcal{P}_{m_{i,t}}}[\|m_{j,t} - v_{i,t,l-1}\|^2 \mathbb{1}\{\|m_{j,t} - v_{i,t,l-1}\| \leq \tau_{i,t,l}\}] \\
&\leq (1+\beta_{i,t,l})c_{i,t,l}^2 + \left(1 + \frac{1}{\beta_{i,t,l}}\right)\tau_{i,t,l}^2 \P_{j:m_{j,t}\nsim\mathcal{P}_{m_{i,t}}}(\|m_{j,t} - v_{i,t,l}\| \leq \tau_{i,t,l})
\end{aligned}
$$

364   If $\|m_{j,t} - v_{i,t,l}\| \leq \tau_{i,t,l}$, then

$$
\begin{aligned}
\Delta^2 \leq \|\mathbb{E}m_{j,t} - \mathbb{E}m_{i,t}\|^2 &\leq 3(\|m_{j,t} - \mathbb{E}m_{j,t}\|^2 + \|m_{j,t} - \mathbb{E}v_{i,t,l-1}\|^2 + \|\mathbb{E}v_{i,t,l-1} - \mathbb{E}m_{i,t}\|^2) \\
&\leq 3(\|m_{j,t} - \mathbb{E}m_{j,t}\|^2 + 2\|m_{j,t} - \mathbb{E}m_{j,t}\|^2 + 2\|\mathbb{E}m_{j,t} - \mathbb{E}v_{i,t,l-1}\|^2 \\
&\quad + 2\mathbb{E}\|v_{i,t,l-1} - \mathbb{E}m_{i,t}\|^2 + 2\mathbb{E}\|m_{i,t} - \mathbb{E}m_{i,t}\|^2) \\
&\leq 3(3\|m_{j,t} - \mathbb{E}m_{j,t}\|^2 + 2\tau_{i,t,l}^2 + 2c_{i,t,l}^2 + 2\rho^2).
\end{aligned}
$$

365   The probability of this event is

$$
\begin{aligned}
\mathbb{P}\left(\|m_{j,t} - \mathbb{E}m_{j,t}\|^2 \geq \frac{\Delta^2}{9} - \frac{2(\tau_{i,t,l}^2 + c_{i,t,l}^2 + \rho^2)}{3}\right) &\leq \frac{\mathbb{E}\|m_{j,t} - \mathbb{E}m_{j,t}\|^2}{\frac{\Delta^2}{9} - \frac{2(\tau_{i,t,l}^2 + c_{i,t,l}^2 + \rho^2)}{3}} \\
&\leq \frac{\rho^2}{\frac{\Delta^2}{9} - \frac{2(\tau_{i,t,l}^2 + c_{i,t,l}^2 + \rho^2)}{3}} \\
&\leq \frac{18\rho^2}{\Delta^2}.
\end{aligned}
$$

366   Therefore

$$
\begin{aligned}
\mathbb{E}_{j:m_{j,t}\nsim\mathcal{P}_{m_{i,t}}}\|y_{i,j,t,l} - \mathbb{E}m_{i,t}\|^2 &\leq (1+\beta_{i,t,l})c_{i,t,l}^2 + \left(1 + \frac{1}{\beta_{i,t,l}}\right)\tau_{i,t,l}^2 \P_{j:m_{j,t}\nsim\mathcal{P}_{m_{i,t}}}(\|m_{j,t} - v_{i,t,l}\| \leq \tau_{i,t,l}) \\
&\leq (1+\beta_{i,t,l})c_{i,t,l}^2 + \left(1 + \frac{1}{\beta_{i,t,l}}\right)\frac{18\tau_{i,t,l}^2(\rho^2 + c_{i,t,l}^2)\rho^2}{\Delta^2} \\
&\leq (1+\beta_{i,t,l})c_{i,t,l}^2 + \left(1 + \frac{1}{\beta_{i,t,l}}\right)\frac{18\sqrt{\delta_{i,t,l}}(\rho^2 + c_{i,t,l}^2)\rho}{\Delta}.
\end{aligned}
$$

367    Now apply the bounds on $\mathcal{T}_1$, $\mathcal{T}_2$, and $\mathcal{T}_3$ to (13), and set $\gamma_{i,t,l} = \frac{1}{\sqrt{\delta_{i,t,l}}}$ and $\beta_{i,t,l} = \sqrt{\delta_{i,t,l}}$.

$$
\mathbb{E}\|v_{i,t,l} - \mathbb{E}m_{i,t}\|^2
$$

$$
\leq \frac{2n_{i,t,l}^2(1+\gamma_{i,t,l})}{N_{i,t,l}^2}\left[\underbrace{\frac{4(c_{i,t,l}^2+\rho^2)\rho}{\sqrt{\delta_{i,t,l}}\Delta}}_{\mathcal{T}_1} + \frac{1}{n_{i,t,l}}\underbrace{\left(\rho^2 + \frac{12(c_{i,t,l}^2+\rho^2)\rho}{\sqrt{\delta_{i,t,l}}\Delta}\right)}_{\mathcal{T}_2}\right]
$$

$$
+ \frac{\left(1+\frac{1}{\gamma_{i,t,l}}\right)(N_{i,t,l}-n_{i,t,l})^2}{N_{i,t,l}^2}\left[\underbrace{(1+\beta_{i,t,l})c_{i,t,l}^2 + \left(1+\frac{1}{\beta_{i,t,l}}\right)\frac{18\sqrt{\delta_{i,t,l}}(\rho^2+c_{i,t,l}^2)\rho}{\Delta}}_{\mathcal{T}_3}\right]
$$

$$
\leq 2\delta_{i,t,l}\left(1+\frac{1}{\sqrt{\delta_{i,t,l}}}\right)\left[\underbrace{\frac{4(c_{i,t,l}^2+\rho^2)\rho}{\sqrt{\delta_{i,t,l}}\Delta}}_{\mathcal{T}_1} + \frac{1}{n_{i,t,l}}\underbrace{\left(\rho^2 + \frac{12(c_{i,t,l}^2+\rho^2)\rho}{\sqrt{\delta_{i,t,l}}\Delta}\right)}_{\mathcal{T}_2}\right]
$$

$$
+ (1+\sqrt{\delta_{i,t,l}})(1-\sqrt{\delta_{i,t,l}})^2\left[\underbrace{(1+\sqrt{\delta_{i,t,l}})c_{i,t,l}^2 + \left(1+\frac{1}{\sqrt{\delta_{i,t,l}}}\right)\frac{18\sqrt{\delta_{i,t,l}}(\rho^2+c_{i,t,l}^2)\rho}{\Delta}}_{\mathcal{T}_3}\right]
$$

$$
\leq 2(\delta_{i,t,l}+\sqrt{\delta_{i,t,l}})\left[\underbrace{\frac{4(c_{i,t,l}^2+\rho^2)\rho}{\sqrt{\delta_{i,t,l}}\Delta}}_{\mathcal{T}_1} + \frac{1}{n_{i,t,l}}\underbrace{\left(\rho^2 + \frac{12(c_{i,t,l}^2+\rho^2)\rho}{\sqrt{\delta_{i,t,l}}\Delta}\right)}_{\mathcal{T}_2}\right] + (1-\delta_{i,t,l})\left[c_{i,t,l}^2 + \frac{18(c_{i,t,l}^2+\rho^2)\rho}{\Delta}\right]
$$

$$
\leq \left[\left(34 + \frac{48}{n_{i,t,l}}\right)\frac{\rho}{\Delta} + (1-\delta_{i,t,l})\right]c_{i,t,l}^2 + \left[\left(34 + \frac{48}{n_{i,t,l}}\right)\frac{\rho}{\Delta} + \frac{4}{n_{i,t,l}}\right]\rho^2.
$$

368    Now set $\Delta$ such that the $\frac{\rho}{\Delta}$ coefficient of $c_{i,t,l}^2$ is bounded above by $\frac{\delta_{i,t,l}}{2}$. This way, the entire
369    coefficient of $c_{i,t,l}^2$ will be bounded above by $1 - \frac{\delta_{i,t,l}}{2}$.

$$
\Delta > \max_{i\in[N],t\in[T],l\in[M_t]} \frac{2\left(34+\frac{48}{n_{i,t,l}}\right)\rho}{\delta_{i,t,l}}.
$$

370    Then,

$$
\mathbb{E}\|v_{i,t,l} - \mathbb{E}m_{i,t}\|^2 \leq \left(1 - \frac{\delta_{i,t,l}}{2}\right)c_{i,t,l}^2 + \left(\left(34+\frac{48}{n_{i,t,l}}\right)\frac{\rho}{\Delta} + \frac{4}{n_{i,t,l}}\right)\rho^2.
$$

$$
(14)
$$

371    Set $c_{i,t,l+1}^2$ to the right side of (14). Unrolling the recursion over $l$ rounds, and remembering that
372    $c_{i,t,0}^2 = 0$,

$$
\mathbb{E}\|v_{i,t,l} - \mathbb{E}m_{i,t}\|^2 \leq \left[\left(\left(34+\frac{48}{n_{i,t,l}}\right)\frac{\rho}{\Delta} + \frac{4}{n_{i,t,l}}\right)\rho^2\right]\sum_{q=0}^{l-1}\left(1-\frac{\delta_{i,t,l}}{2}\right)^q
$$

$$
\leq C\left(\left(34+\frac{48}{n_i}\right)\frac{\rho}{\Delta} + \frac{4}{n_i}\right)\rho^2,
$$

373    where $C$ is a constant to reflect that the series above converges, and $n_{i,t,l}$ is within a constant factor
374    of $n_i$. Therefore, we can conclude that after a single round of thresholding (i.e. $l = 1$),

$$
\mathbb{E}\|v_{i,t,l} - \mathbb{E}m_{i,t}\|^2 \leq C\left(\left(34+\frac{48}{n_i}\right)\frac{\rho}{\Delta} + \frac{4}{n_i}\right)\rho^2. \tag{15}
$$

375    $\qquad\qquad\qquad\qquad\qquad\qquad\qquad\qquad\qquad\qquad\qquad\qquad\qquad\qquad\qquad\qquad\qquad\qquad\square$

 **Convergence of PDL**

 *Proof.* Define $B_i$ to be the RHS of (15), and assume the learning rate $\eta \leq \frac{1}{L}$. For client $i$, by
 $L$-smoothness of $f_i$,

$$f_i(x_{i,t}) \leq f_i(x_{i,t-1}) + \langle \nabla f_i(x_{i,t-1}), x_{i,t} - x_{i,t-1} \rangle + \frac{L}{2}\|x_{i,t} - x_{i,t-1}\|^2$$

$$= f_i(x_{i,t-1}) - \eta\langle \nabla f_i(x_{i,t-1}), v_{i,t,M_t} \rangle + \frac{L\eta^2}{2}\|v_{i,t,M_t}\|^2$$

$$= f_i(x_{i,t-1}) + \frac{\eta}{2}\|v_{i,t,M_t} - \nabla f_i(x_{i,t-1})\|^2 - \frac{\eta}{2}\|\nabla f_i(x_{i,t-1})\|^2 - \frac{\eta}{2}(1 - L\eta)\|v_{i,t,M_t}\|^2$$

$$\leq f_i(x_{i,t-1}) + \eta\|v_{i,t,M_t} - \mathbb{E}m_{i,t}\|^2 + \eta\|\mathbb{E}m_{i,t} - \nabla f_i(x_{i,t-1})\|^2 - \frac{\eta}{2}\|\nabla f_i(x_{i,t-1})\|^2 - \frac{\eta}{2}(1 - L\eta)\|v_{i,t,M_t}\|^2.$$
(16)

 Define $\phi_{i,t} = \mathbb{E}m_{i,t} - \nabla f_i(x_{i,t-1})$. Setting $\alpha \geq 3L\eta$,

$$\|\phi_{i,t}\|^2 = (1 - \alpha)^2\|\mathbb{E}m_{i,t-1} - \nabla f_i(x_{i,t-1})\|^2$$

$$\leq (1 - \alpha)^2(1 + \alpha)\|\mathbb{E}m_{i,t-1} - \nabla f_i(x_{i,t-2})\|^2 + (1 - \alpha)^2\left(1 + \frac{1}{\alpha}\right)\|\nabla f_i(x_{i,t-1}) - \nabla f_i(x_{i,t-2})\|^2$$

$$\leq (1 - \alpha)^2(1 + \alpha)\|\mathbb{E}m_{i,t-1} - \nabla f_i(x_{i,t-2})\|^2 + L^2(1 - \alpha)\left(1 + \frac{1}{\alpha}\right)\|x_{i,t-1} - x_{i,t-2}\|^2$$

$$\leq (1 - \alpha)^2(1 + \alpha)\|\mathbb{E}m_{i,t-1} - \nabla f_i(x_{i,t-2})\|^2 + L^2\eta^2(1 - \alpha)\left(1 + \frac{1}{\alpha}\right)\|v_{i,t-1,M_{t-1}}\|^2$$

$$\leq (1 - \alpha)\|\mathbb{E}m_{i,t-1} - \nabla f_i(x_{i,t-2})\|^2 + \frac{L\eta}{2}(1 - L\eta)\|v_{i,t-1,M_{t-1}}\|^2$$

$$\leq (1 - \alpha)\|\phi_{i,t-1}\|^2 + \frac{L\eta}{2}(1 - L\eta)\|v_{i,t-1,M_{t-1}}\|^2.$$
(17)

 Scale (16) by $L$, take its expectation, and add it to the expectation of (17). Then by Lemma 2,

$$\underbrace{L\mathbb{E}(f_i(x_{i,t}) - f_i(x^*)) + \frac{L\eta}{2}(1 - L\eta)\mathbb{E}\|v_{i,t,M_t}\|^2 + (1 - L\eta)\|\phi_{i,t}\|^2}_{\xi_{i,t}} + \frac{L\eta}{2}\mathbb{E}\|\nabla f_i(x_{i,t-1})\|^2$$

$$\leq L\mathbb{E}(f_i(x_{i,t-1}) - f_i(x^*)) + \frac{L\eta}{2}(1 - L\eta)\mathbb{E}\|v_{i,t-1,M_{t-1}}\|^2 + (1 - \alpha)\|\phi_{i,t-1}\|^2 + L\eta B_i$$

$$\leq \underbrace{L\mathbb{E}(f_i(x_{i,t-1}) - f_i(x^*)) + \frac{L\eta}{2}(1 - L\eta)\mathbb{E}\|v_{i,t-1,M_{t-1}}\|^2 + (1 - L\eta)\|\phi_{i,t-1}\|^2}_{\xi_{i,t-1}} + L\eta B_i.$$
(18)

 Summing (18) over $t$ and observing that $\|\phi_{i,1}\| = \|\mathbb{E}m_{i,1} - \nabla f_i(x_{i,0})\| = 0$,

$$\frac{1}{T}\sum_{t=1}^{T}\frac{L\eta}{2}\mathbb{E}\|\nabla f_i(x_{i,t-1})\|^2 = \frac{1}{T}\left(\sum_{t=2}^{T}\frac{L\eta}{2}\mathbb{E}\|\nabla f_i(x_{i,t-1})\|^2 + \frac{L\eta}{2}\mathbb{E}\|\nabla f_i(x_{i,0})\|^2\right)$$

$$\leq \frac{1}{T}\left(\sum_{t=2}^{T}(\xi_{i,t-1} - \xi_{i,t}) + L\eta B_i\right.$$

$$+ L\mathbb{E}((f_i(x_{i,0}) - f_i(x^*)) - (f_i(x_{i,1}) - f_i(x^*))) + L\eta B_i$$

$$\left. + L\eta\|\mathbb{E}m_{i,1} - \nabla f_i(x_{i,0})\|^2 - \frac{L\eta}{2}(1 - L\eta)\mathbb{E}\|v_{i,1,M_1}\|^2\right)$$

$$\leq \frac{L(f_i(x_{i,0}) - f_i(x^*))}{T} + 2L\eta B_i.$$

Equivalently, setting $\alpha = 3L\eta$

$$\frac{1}{T}\sum_{t=1}^{T}\mathbb{E}\|\nabla f_i(x_{i,t-1})\|^2 \leq \frac{2(f_i(x_{i,0}) - f_i^*)}{\eta T} + 4B_i$$

$$\leq \frac{2(f_i(x_{i,0}) - f_i^*)}{\eta T} + 4C\left(\left(34 + \frac{48}{n_i}\right)\frac{\rho}{\Delta} + \frac{4}{n_i}\right)\rho^2$$

$$\leq \frac{2(f_i(x_{i,0}) - f_i^*)}{\eta T} + 4C\left(34 + \frac{48}{n_i}\right)\frac{\rho^3}{\Delta} + \frac{16C\rho^2}{n_i}$$

$$\leq \frac{2(f_i(x_{i,0}) - f_i^*)}{\eta T} + 4C\left(34 + \frac{48}{n_i}\right)\frac{\alpha^{\frac{3}{2}}\sigma^3}{\Delta} + \frac{16C\alpha\sigma^2}{n_i}$$

$$\leq \frac{2(f_i(x_{i,0}) - f_i^*)}{\eta T} + 4C\left(34 + \frac{48}{n_i}\right)\frac{(3L\eta)^{\frac{3}{2}}\sigma^3}{\Delta} + \frac{48CL\eta\sigma^2}{n_i}$$

$$\leq \frac{2(f_i(x_{i,0}) - f_i^*)}{\eta T} + 1968(L\eta)^{\frac{3}{2}}\frac{\sigma^3}{\Delta} + \frac{48CL\eta\sigma^2}{n_i}.$$

Choose

$$\eta = \min\left\{\frac{1}{3L},\ \left(\frac{2(f_i(x_{i,0}) - f_i^*)}{\frac{48CL\sigma^2}{n_i}T}\right)^{\frac{1}{2}},\ \left(\frac{2(f_i(x_{i,0}) - f_i^*)}{\frac{1968L^{1.5}\sigma^3}{\Delta}T}\right)^{\frac{2}{5}}\right\}.$$

Then

$$\frac{1}{T}\sum_{t=1}^{T}\mathbb{E}\|\nabla f_i(x_{i,t-1})\|^2 \leq 2\sqrt{\frac{96CL(f_i(x_{i,0}) - f_i^*)\sigma^2}{n_iT}}$$

$$+ \left(\frac{1968L^{\frac{3}{2}}\sigma^3}{\Delta}\right)^{\frac{1}{5}}\left(\frac{2(f_i(x_{i,0}) - f_i^*)}{T}\right)^{\frac{4}{5}}$$

$$+ \left(\frac{1968L^{\frac{3}{2}}\sigma^3}{\Delta}\right)^{\frac{2}{5}}\left(\frac{2(f_i(x_{i,0}) - f_i^*)}{T}\right)^{\frac{3}{5}}$$

$$+ \frac{6L(f_i(x_{i,0} - f_i^*)}{T}.$$

$\square$

## A.2   Proof of Theorem 1

See proof of Lemma 2, replacing $\rho$ with $\sigma$, and adding the assumption that, in expectation, cluster-center initializations are $\sigma^2$ close to the true cluster means: $\mathbb{E}\|v_{k_i,0} - \mathbb{E}x_i\|^2 \leq \sigma^2$.

## A.3   Proof of Theorem 2

*Proof.* Let

$$\mathcal{D}_1 = \begin{cases} \delta & \text{w.p. } p \\ 0 & \text{w.p. } 1-p \end{cases}$$

and

$$\mathcal{D}_2 = \begin{cases} \delta & \text{w.p. } 1-p \\ 0 & \text{w.p. } p \end{cases}$$

and define the mixture $\mathcal{M} = \frac{1}{2}\mathcal{D}_1 + \frac{1}{2}\mathcal{D}_2$. Also consider the mixture $\tilde{\mathcal{M}} = \frac{1}{2}\tilde{\mathcal{D}}_1 + \frac{1}{2}\tilde{\mathcal{D}}_2$, where $\tilde{\mathcal{D}}_1 = 0$ and $\tilde{\mathcal{D}}_2 = \delta$. It is impossible to distinguish whether a sample comes from $\mathcal{M}$ or $\tilde{\mathcal{M}}$. Therefore, if you at least know a sample came from either $\mathcal{M}$ or $\tilde{\mathcal{M}}$ but not which one, the best you can do is to estimate $\mu_1$ with $\hat{\mu}_1 = \frac{\delta p}{2}$, half-way between the mean of $\mathcal{D}_1$, which is $\delta p$, and the mean of $\tilde{\mathcal{D}}_1$, which is $0$. In this case

$$\mathbb{E}\|\hat{\mu}_1 - \mu_1\|^2 = \frac{\delta^2 p^2}{4}.$$

If $p \leq \frac{1}{2}$, then
$$\Delta = (1-p)\delta - p\delta = (1-2p)\delta. \tag{19}$$

Also,
$$\rho^2 = \delta^2 p(1-p). \tag{20}$$

Equating $\delta^2$ in (19) and (20),
$$\frac{\Delta^2}{(1-2p)^2} = \frac{\rho^2}{p(1-p)},$$

which can be rearranged to
$$(4\rho^2 + \Delta^2)p^2 - (4\rho^2 + \Delta^2)p + \rho^2 = 0.$$

Solving for $p$,
$$p = \frac{1}{2} - \frac{\Delta}{2\sqrt{4\rho^2 + \Delta^2}}. \tag{21}$$

Note that,
$$\begin{aligned}
\frac{\delta^2 p^2}{4} &= \frac{\rho^2 p^2}{4p(1-p)} \\
&= \frac{\rho^2 p}{4(1-p)}. 
\end{aligned} \tag{22}$$

Plugging the expression for $p$ from (21) into (22), we can see that
$$\frac{\delta^2 p^2}{4} = \frac{\rho^2}{4}\left(\frac{\sqrt{4\rho^2 + \Delta^2} - \Delta}{\Delta}\right) = \frac{\rho^2}{4}\left(\sqrt{1 + \frac{4\rho^2}{\Delta^2}} - 1\right) \geq \frac{\rho^2}{4}\left(\frac{2\rho^2}{\Delta^2} - \frac{2\rho^4}{\Delta^4}\right).$$

The last step used an immediately verifiable inequality that $\sqrt{1+x} \geq 1 + \frac{x}{2} - \frac{x^2}{8}$ for all $x \in [0,8]$.
Finally, we can choose $\Delta^2 \geq 2\rho^2$ to give the result that
$$\mathbb{E}\|\hat{\mu}_1 - \mu_1\|^2 \geq \frac{\delta^2 p^2}{4} \geq \frac{\rho^4}{4\Delta^2}.$$

Finally, suppose that there is only a single cluster with $K = 1$. Then, given $n$ stochastic samples. standard information theoretic lower bounds show that we will have an error at least
$$\mathbb{E}\|\hat{\mu}_1 - \mu_1\|^2 \geq \frac{\rho^2}{4n}.$$

Combining these two lower bounds yields the proof of the theorem. $\qquad\square$

