# OpenReview forum: "Towards Provably Personalized Federated Learning via Threshold-Clustering of Similar Clients"
_NeurIPS.cc/2022/Workshop/Federated_Learning — FL-NeurIPS 2022 Poster_

### Official Review · Reviewer_4A5B · 2022-10-14

Summary:
This paper tackles the Personalized Federated Learning (PFL) problem, in which N agents wish to minimize their own local function (personalized model) as opposed to the classical minimization of the average functions. The authors make the assumption that agents form clusters, and that two agents in the same cluster have the same objective function. Under this cluster assumption, the authors propose an algorithm based on threshold clustering to compute the mean of vectors using only stochastic estimates (this corresponds to the case $f_i(x)=\|x-v_{k_i}\|^2$ where $v_{k_i}$ is the true mean). This threshold clustering approach is then generalized to the optimization case (smooth non-convex setting), where the threshold-clustering is made on ‘‘momentums'', assumed to be different between clusters for the analysis. Experiments are also provided.

Clarity:
The assumptions, theorems and algorithms are clear and well-presented. Theorems 3 and 4 may benefit from a $\mathcal{O} $ notation instead of explicit constants.

Relation to prior works:
There is a ‘‘related work'' paragraph, but some very closely related references are missing. For instance, studying personalization as a stochastic optimization problem as in this paper is not new: [A] and [B] below do exactly this. These papers are quite recent, so the submission would benefit from adding these and commenting the differences with these concurrent works. For instance, [A] studies the convex version of the problem and generalization properties, under general function dissimilarities at the optimum (and no cluster assumption I think, it looks more like a similarity graph approach), and prove general lower bounds, that I think apply also in the framework of the proposed submission, so mentionning them should be done. See also references therein.

[A] ‘‘Sample Optimality and All-for-all Strategies in Personalized Federated and Collaborative Learning'',
Mathieu Even, Laurent Massoulié, Kevin Scaman (arxiv)
(this seems to be another version of ‘‘On Sample Optimality in Personalized Collaborative and Federated Learning'' in Neurips 2022 that I came accross in the accepted papers list, same authors)

[B] ‘‘Linear Speedup in Personalized Collaborative Learning'', El Mahdi Chayti, Sai Praneeth Karimireddy, Sebastian U. Stich, Nicolas Flammarion, Martin Jaggi (arxiv): weighted gradient averaging and a variance reduced strategy (that requires very strong initialization conditions from what I understant). Their approach however seems to be very costly in terms of communications if all agents want to optimize their own function, as opposed to [A] and the proposed submission.
or
‘‘ Optimization with access to auxiliary information''  El Mahdi Chayti, Sai Praneeth Karimireddy (arxiv preprint also), but I think it is just another version of the same paper [B].

Pros:
- Studying PFL as stochastic optimization seems quite promising (see references above), and is quite natural.
- Theoretical guarantees: upper and lower bounds in the mean-estimation problem, that don't match but are ‘‘similar''. Linear speedup in terms of cluster population in the optimization setting in the statistical regime (asymptotic).

Cons:
- Comparison with similar works, as stated above.
- Assumption 6 seems to be quite strong: setting $\alpha=0$ for instance, it means that gradients of functions from different clusters (call them f and g) are different and well separated at all points. This seems unlikely to me, since it for instance means that either f+\delta<g or g+\delta<f, which would not be the case for most of quadratic regressions. That is why such an assumption should only be made at the optimum I think.
- Dependency in terms of number of clusters and more generally robustness to a non-cluster scenario are a problem: what happens if agents are distributed in a non-clustered fashion, but the algorithm is ran? Does this lead to similar guarantees + an error term ?
- I find that the PFL problem for non-convex optimization is quite unnatural: the question is to minimize the number of sample used to access small generalization error. However, in the non-convex setting but using stochastic optimization methods as here, this only leads to finding close-to-stationary points, and thus does not answer the original question. The issue here is not that no generalization error is provided, but more that the setting considered to provide some would be convex optimization: do the proofs generalize to the convex setting? Since biased gradients are easier to study in the non-convex setting, it may not be so, but that could be interesting.


Conclusion:
There are some weaknesses to this paper that cannot be addressed (the last three bullet points above), while the first one could easily be rectified. Yet, I think that for a FL workshop paper submission, due to the fact that the PFL problem is important and the stochastic optimization approach quite promising, this paper ranges as ‘‘border accept''.

---

### Official Review · Reviewer_aMnL · 2022-10-16
**The paper realizes personalization through grouping clients into several clusters and learning a model per cluster. Further convergence guarantees are provided.**

Strength:
The problem of realizing personalization with clustering under heterogeneous FL setups is interesting. Because the clients within each cluster mimic IIDness and train their corresponding models.

The weakness:
1. The reviewer believes that clustering clients using the parameter updates or local momentums is not very effective in FL setups [1,2,3] due to the stochasticity nature of SGD and its updates.
2. No experiment has been provided to visualize the quality of the formed clusters.
3. The accuracy performance of the proposed method should be compared with the newly proposed state-of-the-art clustering and non-clustering personalized method e.g., [1-2].
4. The paper needs to be improved in terms of the logic flow.

[1] H. Jamali rad et al., "Federated learning with taskonomy for non-iid data", IEEE Transactions on Neural Networks and Learning Systems, 2022.
[2] S. Vahidian, et al. "Efficient Distribution Similarity Identification in Clustered Federated Learningvia Principal Angles Between Client Data Subspaces" arXiv preprint arXiv:2209.10526, 2022

---

### Official Review · Reviewer_1WXT · 2022-10-20

To overcome data non-IIDness in federated learning, clustering of clients based on similarity of their data distribution has been proven effective personalization schema, but existing methods lack rigorous theoretical gunratess. This paper aims at bridging this gap by assuming the stochastic gradients on a client may correspond to one of K distributions and show that using  a simple thresholding based clustering algorithm and  local client momentum it is possible to establish  optimal convergence guarantees. The authors also mention that the proposed algorithms are incentive-compatible and can lead to stable coalitions (from a game-theoretic standpoint).



The paper studies an interesting problem and the analysis has merits. I have a few concerns about the assumptions and obtained results. For example, just looking at the clustering algorithm and its guarantees, it can be deduced the mixture of arbitrary well-separated distributions can be learned in an iterative manner with noisy samples from each cluster (stochastic gradients) and I was left wondering how it compares to bath methods for mixture methods in clustering. Moreover, the optimal convergence rate makes sense in  a two-stage schema (first clustering and then optimizing for each cluster), but when it is interleaved with learning global models, how the error in iterative clustering affects the rate.

Despite the above issues, I think the paper has enough merit to be considered for publication as a workshop paper.

---

### Decision · Program_Chairs · 2022-10-20

Accept (Poster)